# Probabilistic harmonization and annotation of single-cell transcriptomics data with deep generative models

Chenling Xu[1,†] iD, Romain Lopez[2,†] iD, Edouard Mehlman[2,3,†] iD, Jeffrey Regier[4] iD, Michael I Jordan[2,5] iD & Nir Yosef[1,2,6,7,*] iD

## Abstract

As the number of single-cell transcriptomics datasets grows, the natural next step is to integrate the accumulating data to achieve a common ontology of cell types and states. However, it is not straightforward to compare gene expression levels across data-sets and to automatically assign cell type labels in a new dataset based on existing annotations. In this manuscript, we demonstrate that our previously developed method, scVI, provides an effective and fully probabilistic approach for joint representation and analysis of scRNA-seq data, while accounting for uncertainty caused by biological and measurement noise. We also introduce single-cell ANnotation using Variational Inference (scANVI), a semi-supervised variant of scVI designed to leverage existing cell state annotations. We demonstrate that scVI and scANVI compare favorably to state-of-the-art methods for data integration and cell state annotation in terms of accuracy, scalability, and adapt-ability to challenging settings. In contrast to existing methods, scVI and scANVI integrate multiple datasets with a single genera-tive model that can be directly used for downstream tasks, such as differential expression. Both methods are easily accessible through scvi-tools.

**Keywords** scRNA-seq; harmonization; annotation; differential expression; variational inference

**Subject Categories** Chromatin, Transcription & Genomics; Computational Biology

**Mol Syst Biol. (2021) 17: e9620**

## Introduction

Recent technological improvements in microfluidics and low volume sample handling (Tanay & Regev, 2017) have enabled the emergence of single-cell transcriptomics (Macosko *et al*, 2017; Zheng *et al*, 2017) as a popular tool for analyzing biological systems (Patel *et al*, 2014; Gaublomme *et al*, 2015; Semrau *et al*, 2017). This growing popularity along with a continued increase in the scale of the respective assays (Angerer *et al*, 2017) has resulted in massive amounts of publicly available data and motivated large-scale community efforts such as the Human Cell Atlas (Regev *et al*, 2017), Tabula Muris (Schaum *et al*, 2018), and the BRAIN Initiative Cell Census Network (bic, 2018). The next natural step in the evolution of this field is therefore to integrate many available datasets from related tissues or disease models in order to increase statistical robustness (Wen & Tang, 2018), achieve consistency and repro-ducibility among studies (Haghverdi *et al*, 2018; Butler *et al*, 2018), and ultimately converge to a common ontology of cell states and types (Wagner *et al*, 2016; Regev *et al*, 2017).

A fundamental step toward the ideal of a common ontology is data *harmonization*, namely integration of two or more transcrip-tomics datasets into a single dataset on which any downstream anal-ysis can be applied. We use the term harmonization rather than *batch effect correction* in order to emphasize that the input datasets may come from very different sources (*e.g.*, technology, laboratory) and from samples with a different composition of cell types. A wide range of methods have already been developed for this fundamental problem, initially for microarrays and later on for bulk RNA sequencing, such as ComBat (Johnson *et al*, 2007) and limma (Ritchie *et al*, 2015). These approaches mainly rely on generalized linear models, with empirical Bayes shrinkage to avoid over-correc-tion. More recently, similar methods have been proposed specifi-cally for single-cell RNA sequencing (scRNA-seq), such as ZINB-

---

1   Center for Computational Biology, University of California, Berkeley, CA, USA
2   Department of Electrical Engineering and Computer Sciences, University of California, Berkeley, CA, USA
3   Centre de Mathématiques Appliquées École polytechnique, Palaiseau, France
4   Department of Statistics, University of Michigan, Ann Arbor, MI, USA
5   Department of Statistics, University of California, Berkeley, CA, USA
6   Ragon Institute of MGH, MIT and Harvard, Boston, MA, USA
7   Chan-Zuckerberg Biohub Investigator, San Francisco, CA, USA
    *Corresponding author. Tel: +1 617 717 9934; E-mail: niryosef@berkeley.edu
    †These authors contributed equally to this work

WaVE (Risso *et al*, 2018), which explicitly accounts for the over-abundance of zero entries in the data. However, because of their linear assumptions, these approaches may not be appropriate when provided with a heterogeneous sample that includes different cell states, each of which may be associated with a different sample-to-sample bias (Haghverdi *et al*, 2018). With these limitations in mind, the next generation of methods turned to non-linear strategies. Broadly speaking, each of these methods includes a combination of two components: (i) joint factorization of the input matrices (each corresponding to a different dataset) to learn a joint low-dimensional latent representation. This is usually done with well-established numerical methods, such as integrative non-negative matrix factorization (LIGER; Welch *et al*, 2019), singular value decomposition (Scanorama; Hie *et al*, 2019), or canonical correlation analysis (Seurat Alignment; Butler *et al*, 2018); (ii) additional non-linear transformation of the resulting latent representations so as to optimally "align" them onto each other. This is usually done using heuristics, such as alignment of mutual nearest neighbors (MNN; Haghverdi *et al*, 2018, Scanorama (Hie *et al*, 2019) and Seurat Anchors; Stuart *et al*, 2019), dynamic time warping (Seurat Alignment; Butler *et al*, 2018), or quantile normalization (LIGER; Welch *et al*, 2019). While this family of methods has been shown to effectively overlay different datasets, it suffers from two important limitations. First, an explicit alignment procedure may be difficult to tune in a principled manner and consequently result in over-normalization. This is especially relevant when the cell type composition is different between datasets and when technical differences between samples are confounded with biological differences of interest. Second, the alignment is done in an ad hoc manner and lacks probabilistic interpretability. Consequently, the resulting harmonized dataset is of limited use and cannot be directly applied for probabilistic decision-making tasks, for example, differential expression. We further discuss work related to harmonization of scRNA-seq data, as well as machine learning research in domain adaptation (from which most of these methods, including ours, built upon) in Appendix Note A.

Besides harmonization, another important and highly related problem is that of automated *annotation* of cell state. In principle, there are two ways to approach this problem. The first is *ab initio* labeling of cells based on marker genes or gene signatures (DeTomaso & Yosef, 2016; Butler *et al*, 2018; DeTomaso *et al*, 2019). While this approach is intuitive and straightforward, its performance may be affected in the plausible case where marker genes are absent due to limitations in sensitivity. The second approach is to "transfer" annotations between datasets. In the simplest scenario, we have access to one dataset where states have been annotated either *ab initio*, or using additional experimental measurements (e.g., protein expression (Zheng *et al*, 2017; Stoeckius *et al*, 2017) or lineage tracing (Weinreb *et al*, 2020)) and another, unannotated dataset from a similar condition or tissue. The goal is to use the labeled data to derive similar annotations for the second dataset, whenever applicable. This task is often complicated by factors such as differences in technology (e.g., using Smart-Seq2 data to annotate 10x Chromium data), partial overlap in cell type composition (i.e., not all labels should be transferred and not all unannotated cells should be assigned a label), complex organization of the labels (e.g., hierarchy of cell types and subtypes (preprint: Wagner & Yanai, 2018), continuum along

phenotypic or temporal gradients), partial labeling (i.e., only a subset of cells from the "annotated" dataset can be assigned a label confidently), and the need to handle multiple (more than 2) datasets in a principled and scalable manner. One way to address the annotation problem with this approach is learning a classifier (preprint: Wagner & Yanai, 2018; Kiselev *et al*, 2018) in order to predict a fixed stratification of cells. However, this approach might be sensitive to batch effects, which could render a classifier based on a reference dataset less generalizable to an unannotated dataset. Another, more flexible approach is to transfer annotations by first harmonizing the annotated and unannotated datasets, thus also gaining from the benefits of having a single dataset that can be subject to additional, joint, downstream analysis.

In this paper, we propose a strategy to address several of the outstanding hurdles in both of the harmonization and annotation problems. We first demonstrate that single-cell variational inference (scVI) (Lopez *et al*, 2018) a deep generative model we previously developed for probabilistic representation of scRNA-seq data—performs well in both harmonization and harmonization-based annotation, going beyond its previously demonstrated capacity to correct batch effects. We then introduce single-cell ANnotation using Variational Inference (scANVI), a new method that extends scVI and provides a principled way to address the annotation problem probabilistically while leveraging any available label information. Because scANVI is able to model cells with or without label information, it belongs to the category of semi-supervised learning algorithms. This flexible framework of semi-supervised learning can be applied to two main variants of the annotation problem. In the first scenario, we are concerned with a single dataset in which only a subset of cells can be confidently labeled (e.g., based on expression of marker genes) and annotations should then be transferred to other cells, when applicable. In the second scenario, annotated datasets are harmonized with unannotated datasets and then used to assign labels to the unannotated cells.

The inference procedure for both of the scVI and scANVI models relies on neural networks, stochastic optimization and variational inference (Kingma & Welling, 2014; Louizos *et al*, 2016) and scales to large numbers of cells and datasets. Furthermore, both methods provide a complete probabilistic representation of the data, which non-linearly controls not only for sample-to-sample bias but also for other technical factors of variation such as over-dispersion, library size discrepancies and zero inflation. As such, each method provides a single probabilistic model that underlies the harmonized gene expression values (and the cell annotations, for scANVI) and can be used for any type of downstream hypotheses testing. We demonstrate the latter point through a differential expression analysis on harmonized data. Furthermore, through a comprehensive analysis of performance in various aspects of the harmonization and annotation problems and in various scenarios, we demonstrate that scVI and scANVI compare favorably to current state-of-the-art methods.

## Results

In the following, we demonstrate that our framework compares favorably to state-of-the-art methods for the problems of harmonization and annotation in terms of accuracy, scalability, and

adaptability to various settings. The first part of the paper focuses on the harmonization problem and covers a range of scenarios, including harmonization of datasets with varying levels of biological overlap, handling cases where the data are governed by a continuous (e.g., pseudotime) rather than discrete (cell types) form of variation, and processing multiple (> 20) datasets. While we demonstrate that scVI performs well in these scenarios, we also show that the latent space learned by scANVI provides a proper harmonized representation of the input datasets—a property necessary for guaranteeing its performance in the annotation problem.

In the second part of this manuscript, we turn to the annotation problem and study its two main settings, namely transferring labels between datasets and *ab-inito* labeling. In the first setting, we consider the cases of datasets with a complete or partial biological overlap and use both experimentally and computationally derived labels to evaluate our performance. In the second setting, we demonstrate how scANVI can be used effectively to annotate a single dataset by propagating high confidence seed labels (i.e., based on marker genes) and by leveraging a hierarchical structure of cell state annotations. Finally, we demonstrate that the generative models inferred by scANVI and scVI can be directly applied for hypotheses testing, using differential expression as a case study.

### Joint modeling of scRNA-seq datasets

We consider a collection of scRNA-seq datasets (Fig 1A and B). After using a standard heuristic to filter the genes and generate a common (possibly large) gene set of size $G$ (Materials and Methods), we obtain a concatenated dataset that may be represented as a matrix. Individual entries $x_{ng}$ of this matrix measures the expression of gene $g$ in cell $n$. Additionally, we use the integer $s_n$ to denote the dataset of origin for each cell $n$. Finally, a subset of the cells may be associated with a cell state annotation $c_n$, which can describe either discrete cell types or hierarchical cell types. More complex structures over labels such as gradients are left as a future research direction.

Since the problem of data harmonization of single-cell transcriptomics is difficult and can potentially lead to over-correction (Appendix Fig S1; Nygaard *et al*, 2016), we propose a fully generative method as a robust and principled approach to address it. In our previous work (Lopez *et al*, 2018), we built single-cell Variational Inference (scVI), a deep generative model where the expression level $x_{ng}$ is zero-inflated negative binomial (ZINB) when conditioned on the dataset identifier ($s_n$), and two additional latent random variables. The first, which we denote by $l_n$, is a one-dimensional random variable accounting for the variation in capture efficiency and sequencing depth. In practice, we noticed that this random variable is highly correlated to the library size (Lopez *et al*, 2018). The second, which we denote as $z_n$, is a low-dimensional random vector that represents the remaining variability (Fig 1B). This vector is expected to reflect biological differences between cells and can be effectively used for visualization, clustering, pseudotime inference, and other tasks. Since the scVI model explicitly conditions on the dataset identifier (in the sense that it learns a conditional distribution, see Materials and Methods), it provides an effective way of controlling for technical sample-to-sample variability. However, scVI is unsupervised and does not make use of the available annotations $c_n$, which can further guide the inference of

an informative latent representation $z_n$. To this end, we present a more refined hierarchical structure for $z_n$. We draw $z_n$ as a mixture conditioned on the cell annotation $c_n$ and another latent variable $u_n$, accounting for further biological variability within a cell type (Materials and Methods). We name the resulting approach single-cell ANnotation using Variational Inference (scANVI).

The variables $z_n$, inferred either with scVI or scANVI, provide an embedding of all cells in a single, joint latent space. Since this latent space is inferred while controlling for the dataset of origin ($s_n$), it inherently provides a way to address the harmonization problem. The annotation of unlabeled cells can therefore be conducted with scVI using their proximity to annotated cells in the joint latent space (e.g., using majority vote over the $k$-nearest neighbors). The scANVI model provides a more principled way to annotate cells, namely through a Bayesian semi-supervised approach. Once fitted, the model is able to provide posterior estimates for the unobserved cell state $c_n$, which can be particularly useful when labels cannot be entirely trusted. Because the marginal distribution $p(x_{ng}, c_n \mid s_n)$ if $c_n$ observed (resp. $p(x_{ng} \mid s_n)$ otherwise) is not amenable to exact Bayesian computation, posterior inference is intractable. Consequently, we use variational inference parameterized by neural networks to approximate the posterior distribution (Kingma & Welling, 2014; Materials and Methods).

Notably, scANVI and scVI both have a certain number of hyperparameters. In the following evaluations, conducted on different datasets and different scenarios, we use the exact same set of hyperparameters in order to demonstrate that our methods can be applied with a minimal requirement of hyperparameter tuning (Materials and Methods). We provide a robustness study for hyperparameters in the context of harmonization in Appendix Fig S2.

### Datasets

We apply our method on datasets generated by a range of technologies (10x Chromium; Zheng *et al*, 2017, 10x, 2017), plate-based Smart-Seq2 (Picelli *et al*, 2014), Fluidigm C1 (Xin *et al*, 2016), MARSSeq (Jaitin *et al*, 2014), inDrop (Klein *et al*, 2015) and CELSeq2 (Hashimshony *et al*, 2016)), spanning different numbers of cells (from a few thousand to over a hundred thousand cells), and originating from various tissues (mouse bone marrow, human peripheral mononuclear blood cells (PBMCs), human pancreas, human, and mouse brain). Datasets are listed and referenced in Appendix Table S1.

### Harmonizing pairs of datasets with a discrete population structure

We conducted a comparative study of harmonization algorithms on four different instances, each consisting of a pair of datasets. The first pair [PBMC-CITE (Stoeckius *et al*, 2017), PBMC8K (10x, 2017)] represents the simplest case, in which the two datasets come from very similar biological settings (i.e., PBMCs) and are generated by the same technology (i.e., 10x) but in different laboratories (i.e., akin to batch correction). A second scenario is that of similar tissue but different technologies, which we expect to be more challenging as each technology comes with its own characteristics and biases (Ziegenhain *et al*, 2017). For instance, some methods (10x, CELSeq2) profile the end of the transcript and use Unique Molecular

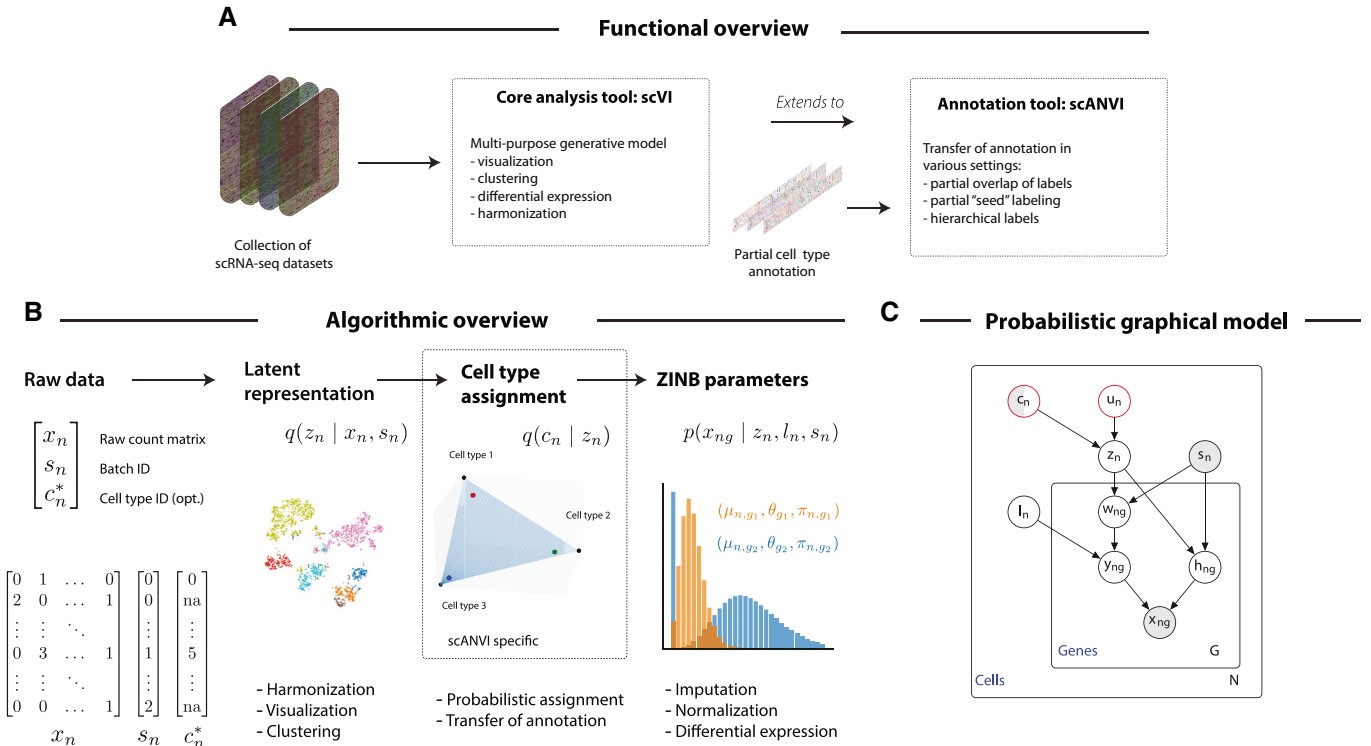

**Figure 1. Harmonization and annotation of scRNA-seq datasets with generative models.**

A  Functional overview of the methods proposed in this manuscript.

B  Schematic diagram of the variational inference procedure in both of the scVI and scANVI models. We show the order in which random variables in the generative model are sampled and how these variables can be used to derive biological insights.

C  The graphical models of scVI and scANVI. Vertices with black edges represent variables in both scVI and scANVI, and vertices with red edges are unique to scANVI. Shaded vertices represent observed random variables. Semi-shaded vertices represent variables that can be either observed or random. Empty vertices represent latent random variables. Edges signify conditional dependency. Rectangles ("plates") represent independent replication. The complete model specification and definition of internal variables is provided in the Materials and Methods

Identifier (UMI) to mitigate inflation in counting, whereas others (e.g., most applications of Smart-Seq2) consider the full length of the transcript without controlling for this potential bias. Additionally, some protocols (e.g., Smart-Seq2) tend to have higher sensitivity and capture more genes per cell compared to others. Finally, studies using droplet-based protocols tend to produce much larger numbers of cells compared to plate-based methods. We explore three such cases, including a bone marrow 10x and Smart-Seq2 pair from the Tabula Muris project (MarrowTM-10x, MarrowTMss2; Schaum *et al*, 2018), a pancreas inDrop and CEL-Seq2 pair (Pancreas-InDrop, PancreasCEL-Seq2; Baron *et al*, 2016), and a dentate gyrus 10x and Fluidigm C1 pair (DentateGyrus10x, DentateGyrus-C1; Hochgerner *et al*, 2018).

Successful harmonization should satisfy two somewhat opposing criteria (Appendix Fig S1). On the one hand, cells from the different datasets should be well mixed; namely, the set of $k$-nearest neighbors ($k$NN) around any given cell (computed e.g., using the euclidean distance in the harmonized latent space) should be balanced across the different datasets. For a fixed value of $k$, this property can be evaluated using the entropy of batch mixing (Haghverdi *et al*, 2018), which is akin to evaluating a simple $k$-nearest neighbors classifier for the batch identifier (Materials and Methods). Briefly, the entropy of batch mixing is the average negative entropy

of batch composition proportion of the $k$-nearest-neighbors of each cell in the harmonized latent space. Higher value for this metric indicates that the harmonized latent space shows strong mixing: the neighbors of each cell are composed of cells from different batches. While this property is important, it is not sufficient, since it can be achieved by simply randomizing the data. Therefore, in our evaluation, we also consider the extent to which the harmonized data retains the original structure observed with each dataset taken in isolation. Here, we expect that the set of $k$-nearest neighbors of any given cell in its original dataset should remain sufficiently close to that cell after harmonization. We evaluate this property using a measure we call $k$-nearest neighbors purity (Materials and Methods), computed as the average percent overlap of the $k$-nearest-neighbors of each cell before and after harmonization. This metric takes value between 0 and 1 and higher values indicate better retainment of structure. This criteria is important, but is maximized by a trivial approach of simply concatenating the latent spaces. Of course, this will result in poor performance with respect to our first measure. Our evaluation therefore relies on both types of measures, namely mixing of data sets and retainment of the original structure. Since our results depend on the neighborhood size $k$, we consider a range of values—from a high resolution ($k = 10$) to a coarse ($k = 500$) view of the data.

We compare scVI to several methods, including MNN (Haghverdi *et al*, 2018), Seurat Alignment (Butler *et al*, 2018), ComBat (Johnson *et al*, 2007), Harmony (Korsunsky *et al*, 2019), Scanorama (Hie *et al*, 2019) and principal component analysis (PCA). In addition, and in order to compare our methods to unpaired data integration approaches based on generative adversarial networks (Zhu *et al*, 2017), we also tested MAGAN (Amodio & Krishnaswamy, 2018). However, even after manual tuning of the learning rate hyperparameter, the input datasets remain largely unmixed (Appendix Fig S3). This might be due to the fact that MAGAN was not directly applied to harmonize pairs of scRNA-seq datasets and need more tuning to be applicable in that context. For each algorithm and pair of datasets, we report embeddings computed via a Uniform Manifold Approximation and Projection (UMAP; McInnes *et al*, 2018) (Appendix Fig S4–S7) as well as quantitative evaluation metrics (Fig 2). Overall, we observed that scVI compares favorably to the other methods in terms of retainment of the original structure (Fig 2A) and performs well in terms of mixing (Fig 2B) for a wide range of neighborhood sizes and across all dataset pairs. The trade-off of these two aspects of harmonization for a fixed $k$ is shown in Fig 2C, and again scVI and scANVI perform favorably and show up on the top right corner of the scatter plot. scANVI performs slightly better than scVI. Furthermore, because the conservation of $k$-nearest neighbors might be more indicative of a local stability of the algorithm and misses the clustering aspect of the data, we also quantified the conservation of cluster assignments. Toward this end, we used the adjusted Rand index to compare the agreement of a $k$-means clustering algorithm, before and after harmonization (Fig 2D; Appendix Table S2). Reassuringly, our positive results for preservation of the output of a clustering algorithm indicate that scVI and scANVI are also stable with regards to more global aspects of the data.

While scANVI was designed for the problem of cell state annotation, we also wanted to evaluate its ability to harmonize datasets, which can be seen as a prerequisite. To evaluate this, we consider each dataset pair twice, each time using labels from one of the datasets (exploiting the semi-supervision framework of scANVI) and leaving the other one unlabeled. Reassuringly, we found that scANVI is capable of effectively harmonizing the datasets, with a similar performance to that of scVI in terms of entropy of batch mixing and retainment of the original structure (Fig 2). We further explore the performance of scANVI in the annotation problem in the subsequent sections.

**Harmonizing datasets with a different composition of cell types**

One of the primary challenges of the harmonization problem is handling cases in which the cell types present in the input datasets only partially overlap or do no overlap at all. Since this is a plausible scenario in many applications, it is important to account for it and avoid over-normalizing or "forcing" distinct cell populations onto each other. To evaluate this, we performed several stress tests in which we artificially manipulated the composition of cell types in the input datasets prior to harmonization. As our benchmark method, we use Seurat Alignment, which performed better than the remaining benchmark methods in our first round of evaluation (Fig 2).

As a case study, we used a pair of PBMC datasets [PBMC-CITE (Stoeckius *et al*, 2017), PBMC-8K (10x, 2017)] that initially contained a similar composition of immune cell types

(Appendix Table S3). We were first interested in the case of no biological overlap (Fig 3A–D). To test this, for a given cell type $c_0$ (e.g., natural killer cells), we only keep cells of this type in the PBMC-CITE dataset and remove all cells of this type from the PBMC-8K dataset. In Fig 3A and B, we show an example of UMAP visualization of the harmonized data, with natural killer cells as the left out cell type $c_0$. Evidently, when harmonizing the two perturbed datasets with scVI, the natural killer cells appear as a separate cluster and are not wrongly mixed with cells of different types from the other dataset. Conversely, we see a larger extent of mixing in the latent space inferred by Seurat Alignment. A more formal evaluation is provided in Fig 3C and D, which presents our harmonization performance metrics for each cell type averaged across all perturbations (in each perturbation, $c_0$ is set to a different cell type). We also included scANVI with the true number of cell types ($C = 6$) in this analysis, using the cell labels from the PBMC-CITE dataset.

Under the ideal scenario of a successful harmonization, we expect both a low entropy of batch mixing (since the datasets do not overlap) and retainment of the original structure. Evidently, both scVI and scANVI exhibit a consistently low level of batch mixing that is better or comparable to that of Seurat Alignment, while retaining the original structure more accurately.

As an additional scenario, we investigated the case where the input datasets contain a similar set of cell types, with the exception of one cell type that appears in only one of the datasets. To simulate this, for a given cell type $c_0$, we removed cells of this type from the PBMC-8K dataset, and then harmonize the remaining cells with the unaltered PBMC-CITE (which still contains $c_0$). We show an example of UMAP visualization in Fig 3E and F, removing CD4$^+$ T cells from the PBMC-8K dataset. Evidently, in the scVI latent space, the PBMC-CITE "unique" CD4$^+$ T cell population is not wrongly mixed with cells from the perturbed PBMC-8K dataset, but rather appears as a distinct cluster. For a more formal analysis, Fig 3G–I shows the harmonization statistics for perturbing the six major cell types present in the PBMC datasets. As above, we also evaluated scANVI in this context, using the labels from the unperturbed (PBMC-CITE) dataset.

Figure 3G shows that the entropy of batch mixing from the "unique" population (averaging over all six perturbations) is low in all three methods (scVI, scANVI, and Seurat Alignment), with a slight advantage for scVI and scANVI. Figure 3H and I shows the harmonization statistics for each population, averaging over all shared cell types between the two datasets. Evidently, for the populations that are indeed common to the two datasets, scVI and scANVI are capable of mixing them properly, while preserving the original structure, comparing favorably to Seurat Alignment on both measures. Overall, the results of this analysis demonstrate that scVI and scANVI are capable of harmonizing datasets with very different compositions, while not forcing erroneous mixing. These results are consistent with the design of scVI and scANVI, which aim to maximize the likelihood of a joint generative model, without making *a priori* assumptions about the similarity in the composition of the input datasets.

In a similar but more complex experiment, we also study the case when the two datasets both have their own unique cell types but also share several common cell types. Populations unique to each dataset have low mixing (Appendix Fig S8A), especially with scVI and scANVI. Conversely, the shared populations have a

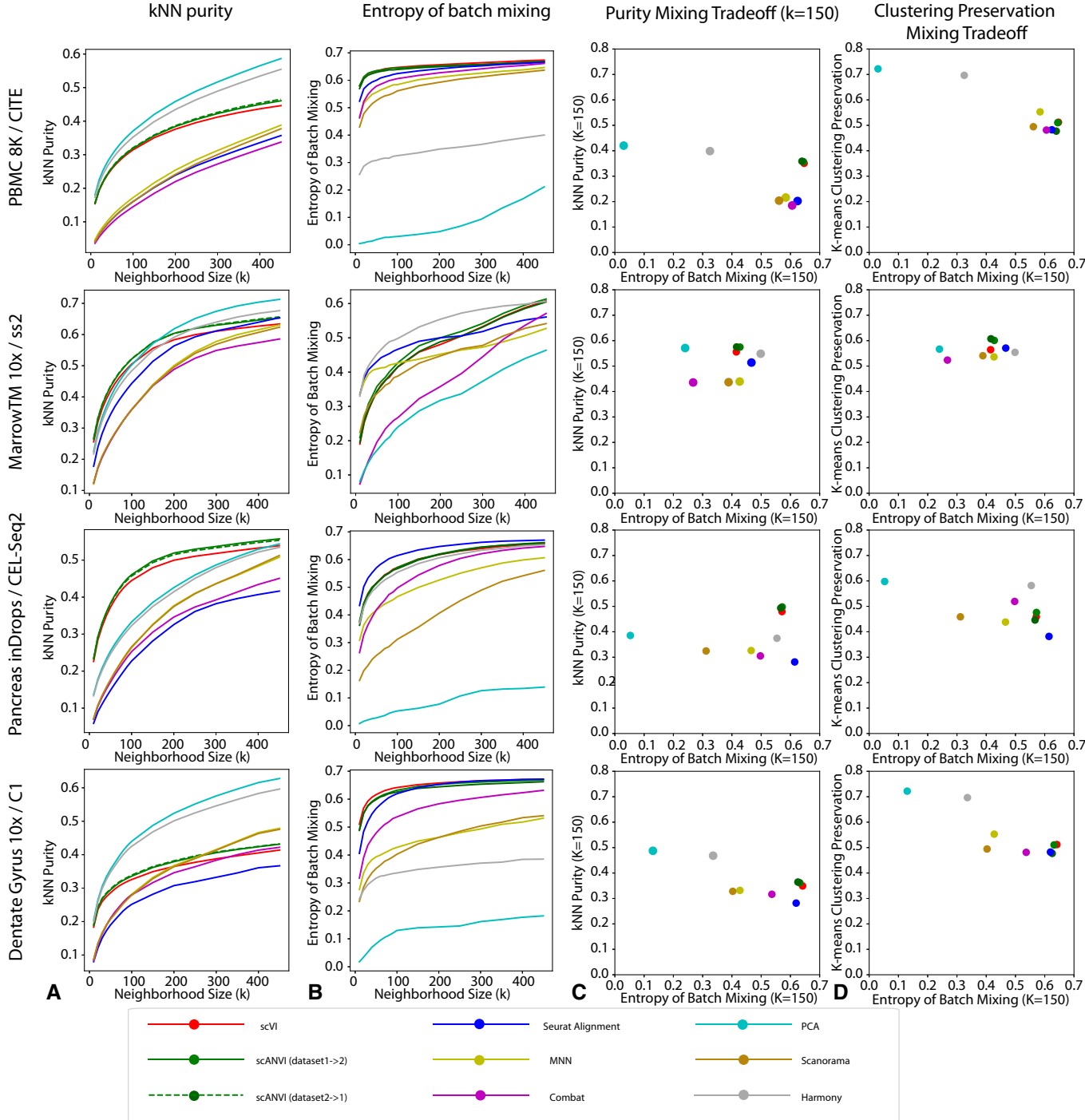

**Figure 2. Benchmarking of scRNA-seq harmonization algorithms.** Each row is a different dataset. Each column is a metric.

A  *k*-nearest neighbors purity that ranges from 0 to 1, with higher values meaning better preservation of neighbor structure in the individual datasets after harmonization.

B  Entropy of batch mixing where higher values means that the cells from different datasets are well mixed.

C  The trade-off between the *k*NN purity and entropy of batch mixing for a fixed *K* = 150. Methods on the top right corner have better performances.

D  The trade-off between entropy of batch mixing and the preservation of biological information using an alternative unsupervised statistic k-means clustering preservation.

substantially higher mixing rate (Appendix Fig S8C). Specifically, scANVI and scVI both mix shared populations better than Seurat, with a better overall performance for scANVI. Finally, the preservation of original structure is higher scVI and scANVI when compared to Seurat across all cell types, especially for B cells, NK cells, and FCGR3A+ Monocytes (Appendix Fig S8B). Overall, these

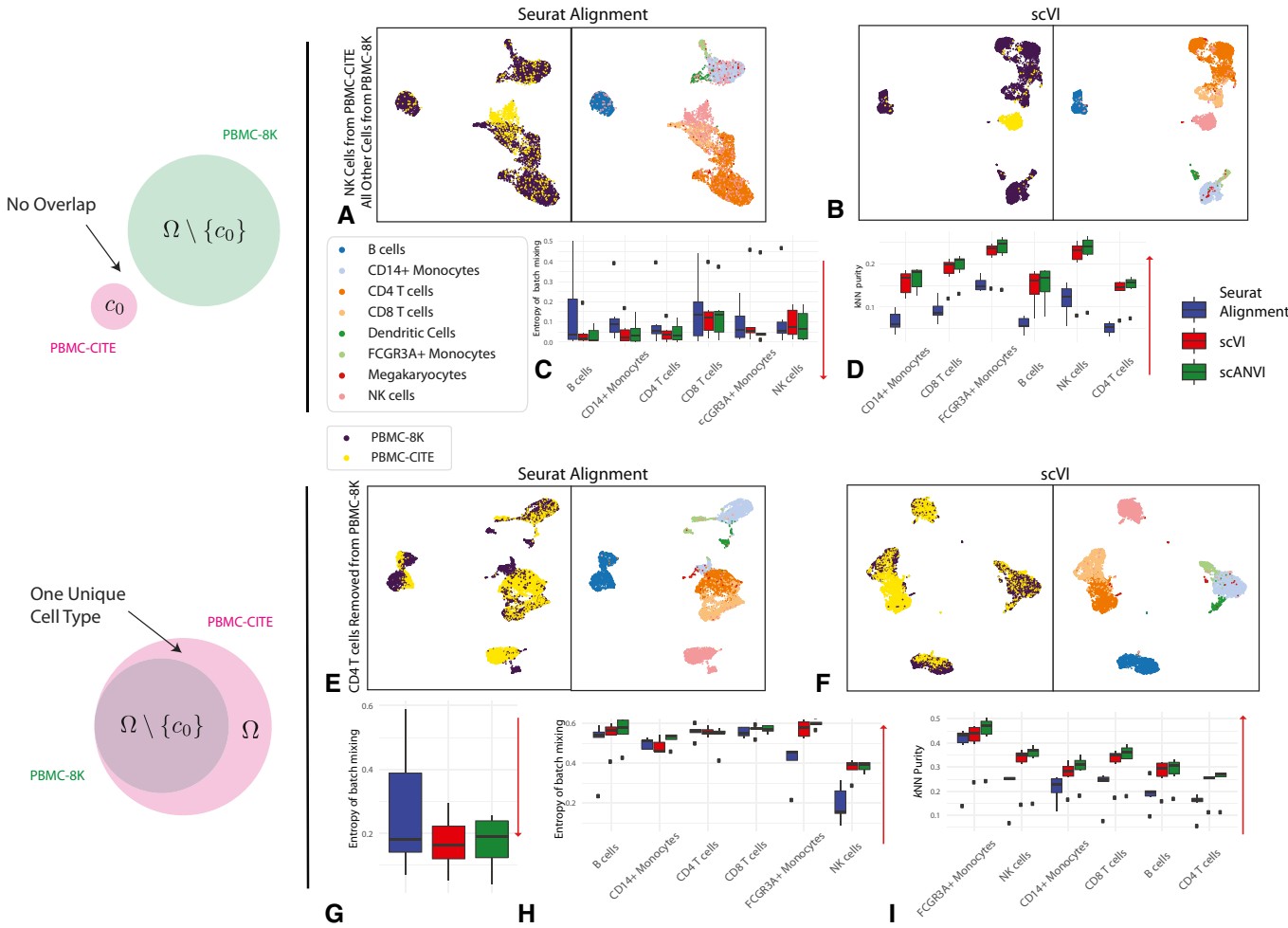

**Figure 3. Harmonizing datasets with different cellular composition.**

A–D The case when no cell type is shared. PBMC-8K contains all cells other than cell type $c_0$ while PBMC-CITE contains only cell type $c_0$. Six experiments were run, each keeping one cell type from the PBMC-CITE dataset. (A, B) UMAP visualization for the case where $c_0$ corresponds to natural killer cells. (C, D) entropy of batch mixing and $k$-nearest neighbors purity, aggregating the six experiments (setting $c_0$ to a different cell type in each experiment). Data information: Red arrows indicate the desired direction for each performance measure. Low batch entropy is desirable in (C) while high $k$-nearest neighbors purity is desirable in (D).

E–I The case when cell type $c_0$ is removed PBMC-8K but not from PBMC-CITE. Six experiments were run, each removing one cell type from the PBMC-CITE dataset. (E, F) UMAP visualization for the case where $c_0$ corresponds to CD4$^+$ T cells. (G) entropy of batch mixing for the removed cell type. Lower value is more desirable as indicated by the red arrow. (H) entropy of batch mixing for the remaining cell types. Higher value is more desirable as indicated by the red arrow. (I) $k$-nearest neighbors purity. Higher value is more desirable as indicated by the red arrow.

results demonstrate that our methods do not tend to force wrong alignment of non-overlapping parts of the input datasets.

**Harmonizing continuous trajectories**

While so far we considered datasets that have a clear stratification of cells into discrete subpopulations, a conceptually more challenging case is harmonizing datasets in which the major source of variation forms a continuum, which inherently calls for accuracy at a higher level of resolution.

To explore this, we use a pair of datasets that provides a snapshot of hematopoiesis in mice [HEMATO-Tusi (Tusi *et al*, 2018), HEMATO-Paul (Paul *et al*, 2015); Fig 4]. These datasets consist of cells along the transition from common myeloid progenitor cells

(Fig 4A and B; middle) through two primary differentiation trajectories myeloblast (top) and erythroblast megakaryocyte (bottom). Notably, the HEMATO-Tusi dataset contains cells that appear to be more terminally differentiated, which are located at the extremes of the two primary branches. This can be discerned by the expression of marker genes (Fig 4E). For instance, the HEMATO-Tusi unique erythroid cell population expresses *Hba-a2* (hemoglobin subunit) and *Alas2* (erythroid-specific mitochondrial 5-aminolevulinate synthase) that are known to be present in reticulocytes (Goh *et al*, 2007, MTA, 2018). At the other end, the granulocyte subset that is captured only by HEMATO-Tusi expresses *Itgam* and *S100a8*. *S100a8* is a neutrophil-specific gene predicted by Nano-dissection (Ju *et al*, 2013) and is associated with GO processes such as leukocyte migration associated with inflammation and neutrophil

aggregation. *Itgam* is not expressed in granulocyte monocyte progenitor cells but is highly expressed in mature monocytes, mature eosinophils, and macrophages (Papatheodorou *et al*, 2017). We therefore do not expect mixing to take place along the entire trajectory. To account for this, we evaluated the extent of batch entropy mixing at different points along the harmonized developmental trajectory. As expected, we find that in most areas of the trajectory the two datasets are well mixed, while at the extremes, the entropy reduces significantly, using either scVI or Seurat Alignment (Fig 4C). Overall, we observe that scVI compares well in terms of both mixing the differentiation trajectories in each dataset and preserving their original, continuous, structure (Fig 4A–D).

To validate scANVI in this context as well, we provided it with the categorical labels of cells along the two developmental trajectories, indicating their cell state (Fig 4C and D and Appendix Fig S9). Even though this labeling scheme does not explicitly account for the ordering between states, we observe that scANVI is capable of mixing the two datasets, while retaining their original structure, achieving a level of accuracy comparable to that of scVI and better than that of Seurat Alignment. We also test the effect of low-quality data in this example where cell types are not clearly demarcated. We observe consistent results, in terms of relative performance between methods, for decreasing rates of sampling in Appendix Fig S10.

### Harmonizing datasets across species

Another more challenging data harmonization scenario is when the two datasets come from different species. Although species share homologous genes, more dataset-specific expression patterns are expected in across-species comparison. We harmonized two datasets from mouse (Saunders *et al*, 2018) and human (Welch *et al*, 2019) Substantia Niagra after mapping homologous genes using the Mouse Genome Informatics Web Site (Bult *et al*, 2019). We visualized the UMAP of the harmonized latent space by scVI and Seurat Alignment (Appendix Fig S11A). Both methods perform well in terms of preserving the cluster structure in the original mouse dataset, as well as mixing the cells from different species. We compare the different harmonization methods more systematically using the *k*NN purity and entropy of batch mixing (Appendix Fig S11B). In this test, we find consistently superior performance of scVI and scANVI.

### Rapid integration of multiple datasets

To demonstrate the scalability of our framework in the context of harmonizing multiple (and possibly large) dataset, we ran scVI to integrate a cohort of 26 datasets spanning 105,476 cells from multiple tissues and technologies, which was made available by the authors of Scanorama (a method based on truncated singular value decomposition followed by nearest neighbor matching (Hie *et al*, 2019)). Using the hardware specified in the original paper (Hie *et al*, 2019) (Intel Xeon E5-2650v3 CPU limited to 10 cores with 384 GB of RAM), Seurat Alignment and MNN required over 24 hours, while Scanorama completed its run in 20 minutes. Using a simpler configuration (eight-core Intel i7-6820HQ CPU with 32 GB RAM) along with one NVIDIA Tesla K80 GPU (GK210GL; addressing 24 GB RAM), we found that scVI integrates all datasets

and learns a common embedding in less than 50 minutes. This running time is competitive considering the reduced memory availability and the increased complexity of our model, compared to that of Scanorama. Notably, all the downstream analyses, such as annotation, differential expression, or visualization can be operated by accessing the latent space or via forward passes through the neural networks. Since these access operations can be conducted very efficiently (Lopez *et al*, 2018), the dominant factor, on which we focused our run time analysis, is the time required for model fitting. Considering the results, the latent space of scVI recapitulates well the major tissues and cell types (Appendix Fig S12), and the position of cells in the latent space provides an effective predictor for the cell type label (Appendix Fig S12 and Materials and Methods).

We also evaluated the runtime of scVI and scANVI on the four smaller dataset pairs we used for benchmarking. We report this metric as a function of the size of the dataset, and compared it to other models used in this paper. The runtime of scVI and scANVI increases as the number of genes increases (Appendix Table S4), but depends largely on the computational resources available at the time, and scales sublinearly. It is thus feasible to run scVI and scANVI with a much larger gene set. However, using more genes does not guarantee better performance, as performance decreases when the number of genes becomes comparable to the number of cells (preprint: Luecken *et al*, 2020).

### Transferring cell type annotations between datasets

We next turned to evaluate scVI and scANVI in the context of harmonization-based annotation. Here, we test the extent to which annotations from a previously annotated dataset can be used to automatically derive annotations in a new unannotated dataset. For scVI and Seurat Alignment, we derive the annotations by first harmonizing the input datasets and then running a *k*-nearest neighbors classifier (setting *k* to 10) on the joint latent space, using the annotated cells to assign labels to the unannotated ones. Conversely, scANVI harmonizes the input datasets while using any amount of available labels. The prediction of unobserved labels is then conducted using the approximate posterior assignments $q_{\Phi}(c \mid x)$ of cell types, directly derived from the model (Materials and Methods). An alternative approach that we benchmark against was taken by scmap-cluster (Kiselev *et al*, 2018). scmap directly builds a classifier based on the labeled cells (instead of performing harmonization first) and then applies this classifier to the unlabeled cells. Finally, we also applied the domain adaptation method Correlation Alignment for Unsupervised Domain Adaptation (CORAL, (Sun *et al*, 2016)). This method was not initially developed for single-cell analysis but is an insightful benchmark from the machine learning literature.

We start by exploring the four dataset pairs in Fig 2, which have been annotated in their respective studies. In each experiment, we "hide" the cell type annotations from one dataset and transfer the second dataset labels to the first one. As a measure of performance, we report the weighted accuracy, which is the percent of cells that were correctly assigned to their correct (hidden) label, averaging over all labels (Materials and Methods). Importantly, the annotations in this first set of case studies were derived computationally. For example, by first clustering the cells, looking for marker genes

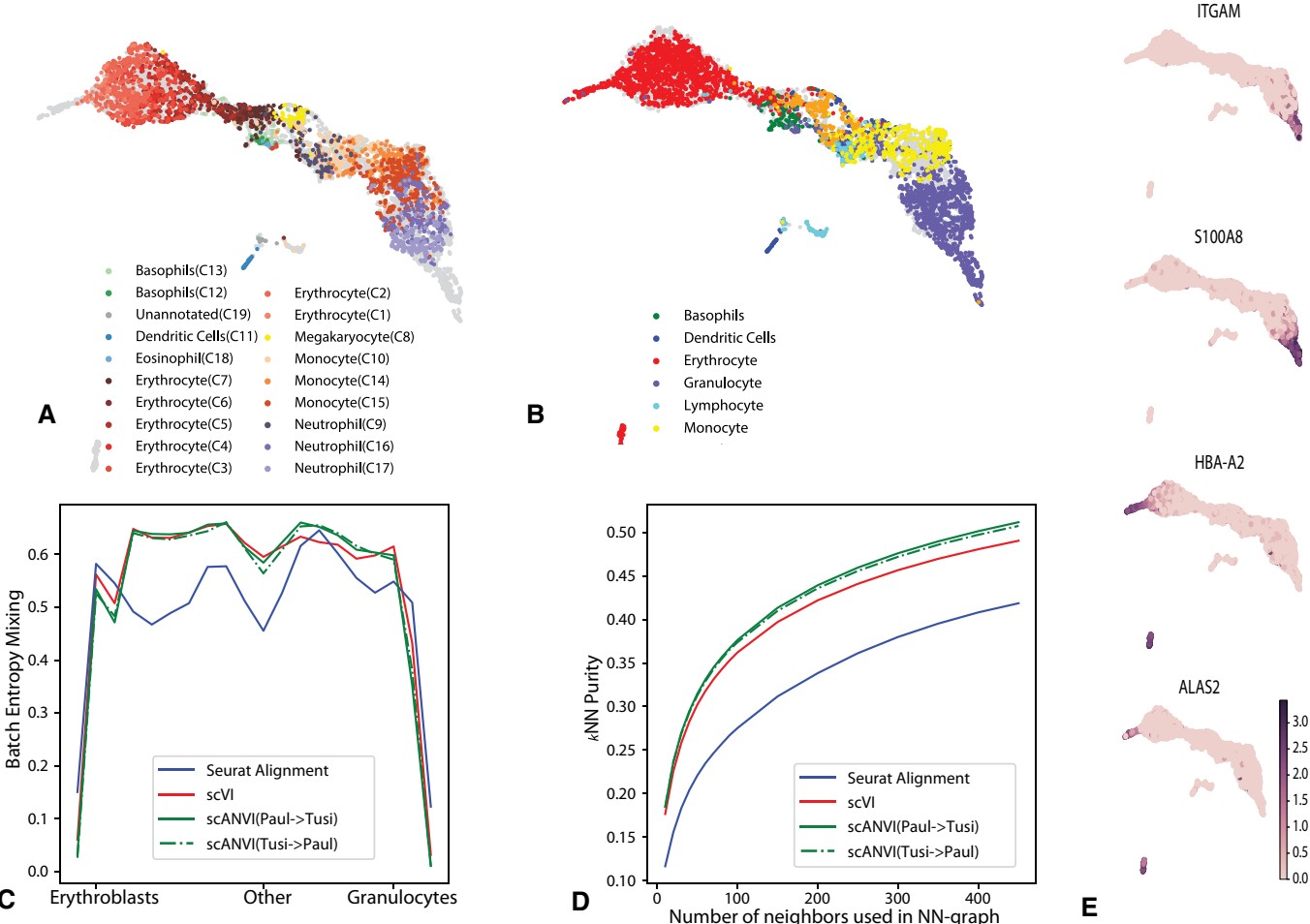

**Figure 4. Harmonizing developmental trajectories.**

A, B  UMAP visualization of the scVI latent space, with cells colored by the original labels from either the HEMATO-Paul (A) or HEMATO-Tusi (B) studies. The cells from the other dataset are colored in gray.

C  Entropy of batch mixing along 20 bins of the HEMATO-Tusi cells, ordered by the potential of each cell. Potential is a pseudotime measure that describes the differentiation state of a cell using the population balance analysis algorithm (center: common myeloid progenitors; moving left: erythrocyte branch; moving right: granulocyte branch).

D  $k$-nearest neighbors purity for scVI, Seurat, and scANVI.

E  Expression of marker genes that help determine the identity of batch-unique cells.

expressed by each cluster and then assigning labels to the clusters accordingly. This level of annotation therefore makes the prediction problem relatively easy, and indeed, while we find that overall scANVI predicts unobserved labels more accurately, the differences between the methods are mild (Appendix Figs S13 and S14). Notably, CORAL achieves overall competitive performance except when transferring labels on the MarrowTM pairs, from 10x to Smart-Seq2. In this specific instance, CORAL maps most of the cells to a single label (incidentally, while this label marks cells that are transcriptionally similar, it is defined by the authors as an unknown class "NA", corresponding to cells that cannot be confidently assigned or low-quality cells according to the authors of (Schaum *et al*, 2018)), which might be due to its linear transformation of the feature space.

To evaluate the accuracy of annotations without the need for computationally derived labels, we turned to the PBMC-CITE dataset which includes measurements of ten key marker proteins in addition to mRNA (Stoeckius *et al*, 2017), and the PBMC-sorted dataset (Zheng *et al*, 2017), where cells were collected from bead purifications for eleven cell types (Appendix Table S5). We applied scVI and scANVI to harmonize and annotate these two datasets along with a third dataset of PBMC (PBMC-68K (Zheng *et al*, 2017)). Our analysis contains a combined set of $n = 169{,}850$ cells from the three datasets altogether. To generate a realistic scenario of cell type annotation, we only provide access to the experimentally based labels from the PBMC-sorted dataset (Fig 5A and B). As an additional benchmark, we also evaluate Seurat Alignment, which was tested after removal of a randomly selected subset (40%) of the two large datasets (PBMC68K and PBMC-sorted) due to scalability issues. Considering our harmonization performance measures (i.e., retainment of the original structure and batch mixing), we observe as before that scVI and scANVI perform

similarly and compare favorably to Seurat Alignment. We then evaluated the accuracy of assigning unobserved labels, focusing on the PBMC-CITE dataset. Instead of using the labels from the original PBMC-CITE study as ground truth (which were computationally derived), we used the protein data, which provides an experimentally derived proxy for cell state. To this end, we quantified the extent to which the similarity between cells in the harmonized mRNA-based latent space is consistent with their similarity at the protein level (Materials and Methods). We first computed the average discrepancy (sum of squared differences) between the protein measurements in each cell and the average over its $k$-nearest neighbors. As a second measure, we computed for each PBMC-CITE cell the overlap between its $k$-nearest PBMC-CITE neighbors in the harmonized mRNA-based space and in the protein space. We then report the average across all cells in Appendix Fig S15. Evidently, scANVI outperformed both scVI and Seurat Alignment for a wide range of neighborhood sizes, providing a representation for the mRNA data that is more consistent with the protein data (Fig 5C).

To provide a more intuitive view of the data we show the level of protein marker measurements on the scVI latent space (Fig 5D) and two examples of mis-annotations clearly visible from our re-analysis (Fig 5E).

### Cell type annotation in a single dataset based on "seed" labels

An important variant of the annotation problem lies within the context of an *ab initio* labeling of a single dataset where only a subset of the cells can be confidently annotated based on the raw data. This increasingly prevalent scenario may result from limited sensitivity of the scRNA-seq assay, where marker genes may only be confidently observed in a small subset of cells. One common way to address this problem is to compute some form of a distance metric between cells (e.g., after embedding with scVI or using Seurat PCA) and then assign labels based on proximity to annotated cells (Zheng *et al*, 2017). To benchmark our methods, we consider two such predictors: The first is clustering the cells and taking a

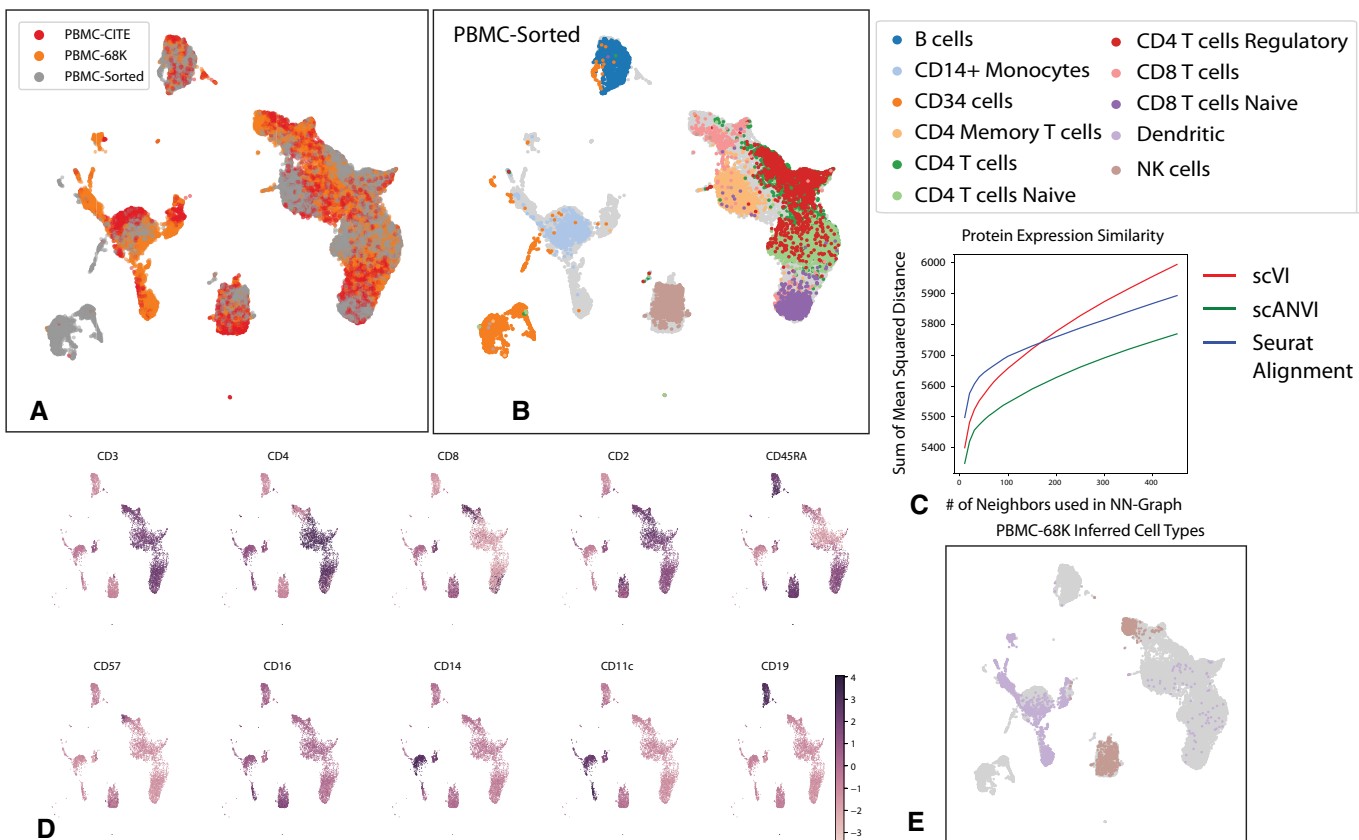

**Figure 5. Validation of cell type annotations using additional metadata.**

A, B UMAP plot of the scANVI latent space inferred for three harmonized datasets: PBMC-CITE, PBMC-sorted, and PBMC-68K. Cells are colored by the dataset of origin (A) and the PBMC-sorted labels (B). Cells from the PBMC-CITE and PBMC-68K are colored in gray in (B).

C The consistency of the harmonized PBMC-CITE mRNA data with the respective protein measurements, evaluated by mean squared error and for different neighborhood size. Lower values indicate higher consistency.

D UMAP plot of the scANVI latent space, where cells are colored by normalized protein measurement. Only PBMC-CITE cells are displayed.

E UMAP plot of the scANVI latent space, with cells from the PBMC-68k dataset colored according to their original label. For clarity of presentation, only cells originally labeled as dendritic cells or natural killer cells are colored. Evidently, a large number of these cells are mapped to a cluster of T cells (right side of the plot).

majority vote inside each cluster, and the second is taking the majority vote of the *k*-nearest neighbors around each unannotated cell ($k = 10$). While these approaches are quite straightforward, their accuracy might suffer when the data do not form clear clusters (Tusi *et al*, 2018), or when differences between labels are too subtle to be captured clearly by a transcriptome-wide similarity measure. To address these issues, scANVI takes an alternative approach, namely learning a latent embedding that is guided by the available labels, and then producing posterior probabilities for assigning labels to each cell.

As a case study, we compiled a dataset consisting of several experimentally sorted and labeled subsets of T cells from the PBMC-sorted dataset, including CD4 memory, CD4 naive, CD4 regulatory, and CD8 naive. To make our analysis more realistic, we assume that the labels are completely unknown to us and therefore assign each T cell to its respective subset using marker genes (12 altogether; see Materials and Methods). Notably, several important biomarkers (*CD4*, *CTLA4*, and *GITR*) are detected in less than 5% of the cells. This renders their use for annotation not straightforward. Furthermore, many of these biomarkers are sparsely expressed to the extent that they are likely to be filtered out in the gene selection step of most harmonization procedures (Fig 6A).

To analyze this dataset, we first computed a signature score for each cell and for each label (i.e., T cell subset) using the scaled raw expression values of the respective marker genes (Materials and

Methods). We then designated the top 50 scoring cells in each subset as the seed set of cells that are confidently annotated for that subset (Fig 6B). Reassuringly, this partial annotation is in agreement with the experimentally derived cell type labels available for this dataset (Fig 6C). However, this dataset does not form clear clusters, and in particular the seed sets of cells are not well separated. Such an observation makes clustering-based approaches potentially less precise. Indeed, using *k*-means clustering on the scVI and Seurat PCA latent space, we find that 74% and 72% of the cells were assigned with their correct label. Similar analysis with two additional popular clustering algorithms (DBSCAN (Ester *et al*, 1996) and PhenoGraph (Levine *et al*, 2015)) further emphasizes the challenge of a cluster-based approach on this data. Specifically, DBSCAN does not partition the data into more than one cluster (scanning through a large number of parameter values; Materials and Methods), and PhenoGraph predicts 9 clusters and achieves an accuracy of 41% (Appendix Fig S16).

Consistent with these results, the application of a *k*-nearest neighbors classifier resulted in a similar level of accuracy in the Seurat PCA latent space (71%), which is slightly improved when replacing it with the scVI latent space (73%; Appendix Fig S16). Conversely, after fitting the scANVI model based on this partial labeling, the annotation posterior $q_{\Phi}(c \mid z)$ (Fig 6d) provides a substantially more accurate cell type assignment, with 84% of cells annotated correctly.

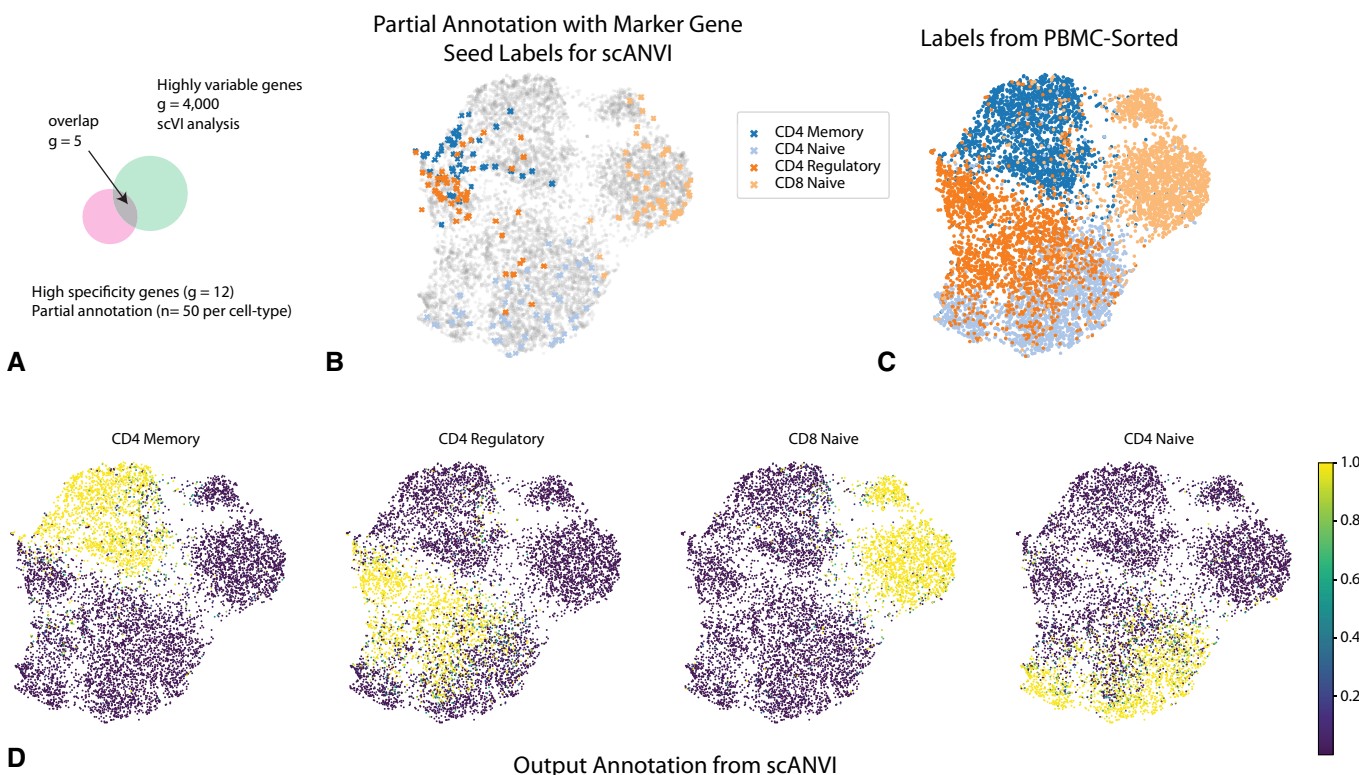

**Figure 6. Cell type annotation in a single dataset using "seed" labeling.**

A    Discrepancies between marker genes that can be used to confidently label cells and highly variable genes in scRNA-seq analysis.

B–D    UMAP plot of the scVI latent space. (B) Seed cells are colored by their annotation (using known marker genes). (C) PBMC-sorted cell type labels from the original study based on marker-based sorting. (D) The posterior probability of each cell being one of the four T cell subtypes obtained with scANVI.

While scANVI has been designed to handle discrete (but not continuous) labels, we hypothesized that gradual transition between cell states may still be captured by the uncertainty of label assignment. We tested it using simulated data (Zhang *et al*, 2019) that consists of a set of "end-point" states along with intermediary states that connect them (Materials and Methods, Appendix Fig S17A). We provided labels only to end-point cells, and investigated the label assignment scores calculated for the intermediary cells. We find that scANVI provides a range of assignment probability values and that these values are proportional to the distance from the respective end-points (Appendix Fig S17B–G). Conversely, the scores provided by scmap tend to be more extreme (Appendix Fig S17H and I), thus less reflecting the continuous nature of the data. This experiment suggests that scVI and scANVI work well with dataset where transcriptional states change gradually and do not have clearly demarcated boundaries. This property could be useful in analyzing other similarly challenging dataset such as tumor samples.

## Cell type taxonomy and hierarchical classification with scANVI

Another subtle yet important variation of the annotation problem is when the labels are not mutually exclusive but rather form a taxonomy of cell types or states. To effectively annotate cells in this setting, we extended scANVI to perform hierarchical classification, which as before we carry out from first principles, relying on probabilistic graphical models (Materials and Methods). To demonstrate this extended version, we use a dataset of the mouse nervous system (Zeisel *et al*, 2018) that was annotated using a cell type taxonomy with several levels of granularity. At the lowest (most granular) level, the cells are stratified into 265 cell subtypes. At the second lowest level of granularity, these 265 subtypes are grouped into 39 subsets, each corresponding to a more coarse definition of a cell type.

We evaluate the ability of scANVI as well as the competing methods at inferring the most granular level of labels when provided with partial "seed" annotation—namely label information for 5 randomly selected cells per label (which accounts for an overall of 0.8% of the cells). We first observe that Seurat PCA followed by a $k$-nearest neighbors classifier provides a weighted accuracy of 23% (averaging over all cell types). While this might seem like a low accuracy, it is in fact far from trivial since the expected weighted accuracy of a random classifier or a constant predictor is of around $1/265 \approx 0.3\%$. Such low numbers are due to the high number of labels at this highly granular scale. scVI provides a substantially better, yet still low level of accuracy at 32%. Interestingly, when scANVI is used without accounting for hierarchy, its performance is similar to the unsupervised scVI (at 32%), which might result from very large number of labels that may require hyperparameter tuning (e.g., increasing the number of classifier training epochs; see Appendix Note B). However, when we take the hierarchy of the labels into account, the performance of scANVI increases to 37%, thus outperforming the other methods by a significant margin. Notably, while we tested the extrapolation of seed labeling and the hierarchical mode only in the context of a single dataset, this variation of the scANVI model can also be directly applied in the context of multiple datasets (i.e., transferring hierarchical annotations between datasets).

## Hypotheses testing in harmonized datasets: the case of differential expression

With their probabilistic representation of the data, scVI and scANVI each provide a natural way of performing various types of hypotheses testing (Materials and Methods). This is different from other approaches (Haghverdi *et al*, 2018; Butler *et al*, 2018; Welch *et al*, 2019; Hie *et al*, 2019; Stuart *et al*, 2019) where the dataset alignment procedures do not carry direct probabilistic interpretation, and the resulting harmonized data can thus not be directly used for these purposes.

To demonstrate this, we focus on the problem of differential expression. As a first case study, we use two of the PBMC datasets (PBMC-8K and PBMC-68K) and looked for differentially expressed genes in two settings: comparing the B cells to dendritic cells, and similarly for CD4$^+$ versus CD8$^+$ T cells. For evaluation, we used reference sets of differentially expressed genes that were obtained from published bulk-level analysis of similar cell subsets (microarrays, (Görgün *et al*, 2005; Nakaya *et al*, 2011), as in (Lopez *et al*, 2018)). While this benchmark relies on real data, a clear caveat is the lack of a well-defined ground truth. To address this, we used a second benchmark based on simulations with Symsim (Zhang *et al*, 2019). The simulated data consists of five subpopulations of varying degrees of transcriptional distance, profiled in two different "batches" of different technical quality (Materials and Methods, Fig S18). This framework allowed us to derive an exact log fold changes (LFC) between every pair of simulated subpopulations, which enable a more accurate evaluation of performance (Fig 7A).

In both benchmark studies, we assume that labels are only available for one of the two input batches or datasets (in the real data we assume that PBMC-8K is the annotated one). To apply scVI, we first harmonized the input pair of datasets and transferred labels using a $k$-nearest neighbors classifier on the joint latent space ($k = 10$). We then consider these annotations (predicted and pre-labeled) as fixed and sample 100 cell pairs, each pair consisting of one cell from each population. For each cell pair, we sample gene expression values from the variational posterior, while marginalizing over the different datasets, to compute the probability for differential expression in a dataset-agnostic manner. Aggregating across all selected pairs results in approximate Bayes factors that reflect the evaluated extent of differential expression (Materials and Methods). Since scANVI assigns posterior probability for associating any cell to any label, it enables a more refined scheme. Specifically, instead of sampling pairs of cells, we are sampling pairs of points in the latent space, while conditioning on the respective label. This approach therefore does not assume a fixed label for each cell (or point in latent space) as in the scVI scheme, but rather a distribution of possible labels thus making it potentially more robust to mislabeling. For reference, we also included edgeR (Robinson *et al*, 2010) using the same labels as scVI. Notably, edgeR was shown to perform well on scRNA-seq data (Soneson & Robinson, 2018) and uses a log-linear model to control for technical sample-to-sample variation.

In our simulations, we considered differential expression between every possible pair out of the five simulated subpopulations. For evaluation, we computed the Spearman and Kendall rank correlation coefficients between the true LFC and the inferred Bayes factors (for scVI and scANVI) or estimated LFC (for edgeR). Our results in Fig 7A show that with this artificial, yet more clearly

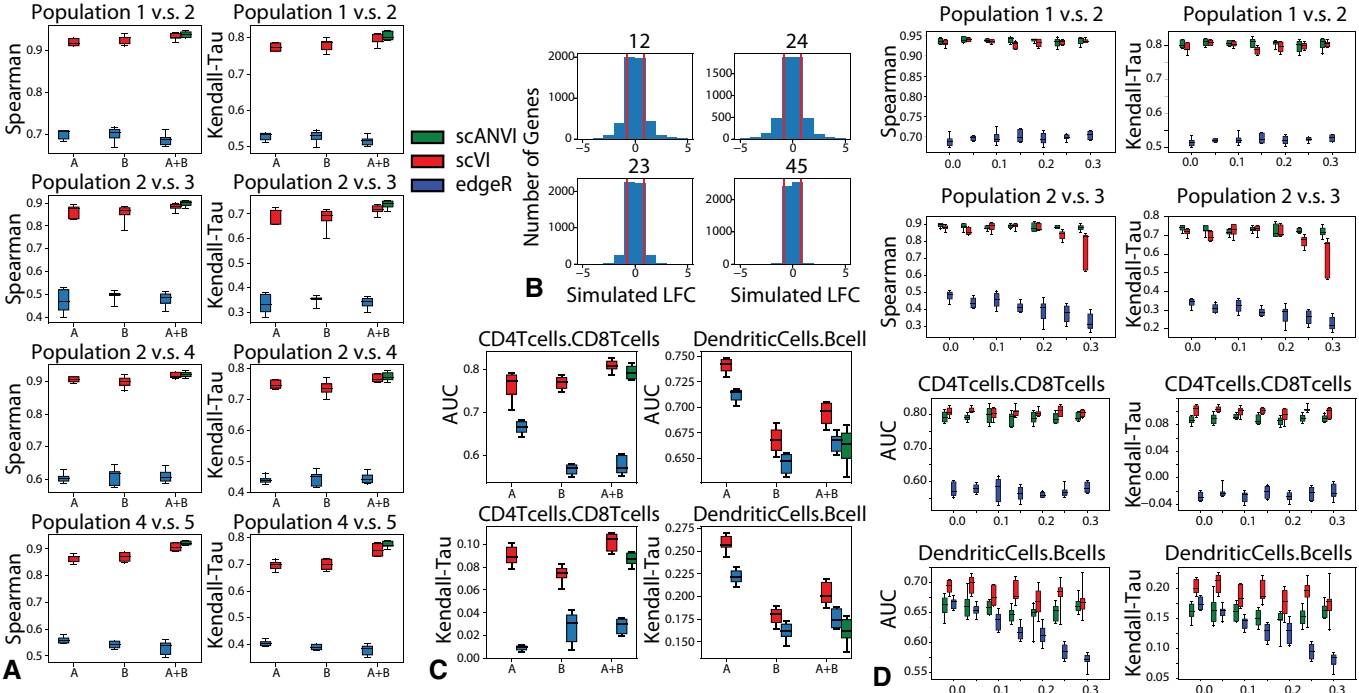

**Figure 7.  Differential Expression on multiple datasets with scVI.**

A   Evaluation of consistency with Spearman rank correlation and Kendall-Tau is shown for comparisons of multiple pairs of cell types in the simulated data. For each comparison, we subsampled 30 cells from each group, and repeated the subsampling 10 times to evaluate the uncertainty in our result.
B   Distribution of true log fold change between all pairs of cell types for the simulated data. The pairs of cells are chosen to represent different levels of distance on the tree as in Appendix Fig S18A. The pairs of population from most distant to least distant are "12", "24", "23", "45".
C   Evaluation of consistency with the AUROC and Kendal Tau metric is shown for comparisons of CD4 vs CD8 T cells and B cells vs dendritic cells on the PBMC-8K only (A), the PBMC-68k only (B) and the merged PBMC-8K / PBMC-68K (A + B) for scVI and edgeR. For each comparison, we subsampled 30 cells from each group, and repeated the subsampling 10 times to evaluate the uncertainty in our result.
D   Mislabeling experiment in differential expression in both the SymSim simulated datasets and in the PBMC8K and PBMC68K dataset. The top row shows differential expression results for the correctly labeled population pair (Population 1 vs. Population 2 in simulated dataset and CD4 T cells vs. CD8 T cells in PBMC dataset). The bottom row shows differential expression results for the mislabelled population pair (Population 2 vs. Population 3 in simulated dataset and dendritic cells vs. B cells in PBMC dataset). For all, x-axis represents the proportion of flipped labels.

Data information: The boxplots are standard Tukey boxplots where the box is delineated by the first and third quartile and the whisker lines are the first and third quartile plus minus 1.5 times the box height. The dots are outliers that fall above or below the whisker lines. The center band indicates the median.

defined objective, scVI was substantially more accurate than edgeR and that in the harmonized data scANVI provided more exact and stable estimates than scVI. The difficulty of each paired comparison is visualized by histograms of the simulated LFC (Fig 7B).

To evaluate performance on the real data, we defined genes as differentially expressed if the adjusted p-value in the reference bulk data (provided by (Görgün *et al*, 2005; Nakaya *et al*, 2011)) was under 5%. Considering these genes as positive instances, we calculated the area under the ROC curve (AUROC) based on rank ordering the inferred Bayes factors (for scVI and scANVI) or p-values (for edgeR). Since the definition of positives genes required a somewhat arbitrary threshold, we also used a second score that evaluates the reproducibility of gene ranking (bulk reference vs. single-cell; considering all genes), using the Kendall rank correlation coefficient (Fig 7C). As a reference, we look at the accuracy of differential expression analysis in each PBMC dataset separately (using their prior annotations to define the sets of cells we are comparing), which can be computed with scVI (as in (Lopez *et al*, 2018)) and edgeR. Reassuringly, we observe that the performance of scVI on the joint

data is not lower than it is in either dataset in isolation. We also find that while scVI performs moderately better than scANVI, both methods compare favorably to edgeR in terms of accuracy.

Mislabeling of a certain proportion of cells in a dataset is a plausible scenario that may occur in any study. An important challenge is therefore to maintain the validity of downstream analysis despite such "upstream" annotation errors. To evaluate robustness in this setting, we repeated the simulation analysis, while introducing labeling errors at different rates. Specifically, prior to evaluating differential expression between two simulated subpopulations, we flip the labels of a certain proportion (up to 30%) of the respective cells in the annotated batch. We then proceed as before and assign labels to cells in the unannotated batch by scVI or scANVI, followed by differential expression analysis. Our results (Fig 7D) suggest that scANVI is clearly more robust to this type of mislabeling than scVI (or edgeR, applied on the scVI- derived labels). Repeating the same analysis on the PBMC data (where the differential expression ground truth is obviously not available), we observe similar level of robustness in scANVI, albeit with not much difference compared to scVI and edgeR.

Overall, our results demonstrate that both scVI and scANVI are capable of conducting differential expression effectively, while working directly on a harmonized dataset. Furthermore, we observe that both methods and especially scANVI are robust to mislabeling, providing further motivation for explicitly modeling label uncertainty.

# Discussion

In this study, we demonstrated that scVI provides a principled approach to harmonization of scRNA-seq data through joint probabilistic representation of multiple dataset, while accounting for technical hurdles such as variable library size and limited sensitivity. We have demonstrated that scVI compares favorably to other methods in its accuracy and that it scales well, not only in terms of the number of cells (as in (Lopez *et al*, 2018)) but also the number of input datasets (as opposed to other methods that work in a pairwise fashion and therefore scale quadratically with dataset size (Hie *et al*, 2019)). We have also shown that the harmonization step of scVI provides an effective baseline for automated transfer of cell type labels, from annotated datasets to new ones.

While the performance of scVI in the annotation problem compares favorably to other algorithms, it does not make use of any existing cell state annotations during model training, but rather after the latent space has been learned. To make better use of these annotations (which may be available for only some of the input datasets or only some cells within a dataset), we developed scANVI, a semi-supervised variant of scVI. While the latent space of scVI is defined by a Gaussian vector with diagonal unit variance, scANVI uses a mixture model, which enables it to directly represent the different cell states (each corresponding to a mixture component; see Materials and Methods) and provide a posterior probability of associating each cell with each label. We have demonstrated that similar to scVI, scANVI is capable of harmonizing datasets effectively. In addition, scANVI provides a way to address a number of variants of the annotation problem. Here, we have first shown that it performs well in the most prevalent application of transferring labels from a reference dataset to an unannotated one. We then demonstrated that scANVI can be used in the context of a single unannotated dataset, where high confidence ("seed") labels are first inferred for a few cells (using marker genes) and then propagated to the remaining cells. Finally, we have shown that scANVI is especially useful in the challenging case where the differences between cell states are too subtle to be captured clearly by a transcriptome-wide similarity measure, as well as in the case where the labels are organized in a hierarchy.

Notably, although scANVI achieves high accuracy when transferring labels fromone dataset to another, it was not designed to automatically identify previously unobserved labels. Indeed, in Appendix Fig S19, we demonstrate that increasing the number of labels in the model (*C*) to values beyond the number of observed labels does not alter the results much. Nevertheless, we observed that unannotated cell populations that have an unobserved label are associated with low levels of mixing between the input datasets. We therefore advocate that clusters from an unannotated dataset that do not mix well should be inspected closely and, if appropriate, should be manually assigned with a new label.

One concern in applying methods based on neural networks (Ding *et al*, 2018; Wang & Gu, 2018; Amodio *et al*, 2019; Eraslan

*et al*, 2019; Grønbech *et al*, 2020) in single-cell genomics and other domains is the robustness to hyperparameters choices (Hu & Greene, 2019). This concern has been addressed to some extent by recent progress in the field, proposing search algorithms based on held-out log-likelihood maximization (Eraslan *et al*, 2019). In this manuscript, we used an alternative approach that is more conducive for direct and easy application of our methods—namely we fix the hyperparameters and achieve state-of-the-art results on a substantial number of datasets and case studies.

The development of scVI and scANVI required several modeling and implementation choices. In Appendix Note C, Appendix Figs S20 and S21, we discuss the rationale behind the choice of a zero-inflated negative binomial (ZINB) distribution as well as robustness to choice of priors. Briefly, we find that exclusion of zero inflation from the model results in approximately similar performance, which is consistent with findings in (Hafemeister & Satija, 2019; Townes *et al*, 2019). The only exception is the case of harmonizing Smart-Seq2 and 10x data sets, in which ZINB performs significantly better. Such results might suggest that zero inflation may be more suitable for certain technologies than others. Similarly, we investigate the prior on the library size which is defined per batch and show that computing the same prior for both the datasets (rather than each dataset individually, as we do by default) affects the performance only in the case of harmonizing the same pair of Smart-Seq2 and 10x datasets. Since these datasets have very different sources of technical noise, this may suggest that it is indeed advisable to explicitly account for such differences during model fit.

An important distinguishing feature of both scVI and scANVI is that they rely on a fully probabilistic model, thus providing a way to directly propagate uncertainties to any downstream analysis. While we have demonstrated this for differential expression analysis and cell type annotation, this can be incorporated to other tasks, such as differential abundance of subpopulations in case-control studies, correlation between genes and more. We therefore expect scVI, scANVI and similar tools to be of much interest as the field moves toward the goal of increasing reproducibility and consistency between studies and converging on to a common ontology of cell types. In particular, we expect scANVI to be especially useful for transferring labels while taking into account the uncertainty, or in the case of a more complex label structure such as hierarchical cell types. Finally, as recent preprints propose proof of concepts for integrating single-cell data across different data modalities such as Single molecule fluorescent in situ hybridization (smFISH), RNA-seq, ATAC-seq, and DNA methylation (Welch *et al*, 2019; Stuart *et al*, 2019), further work can utilize probabilistic graphical models that quantify measurement uncertainties in each assay, as well as the uncertainties of transferring information between modalities (e.g., predicting unmeasured gene expression in smFISH data as in (Lopez *et al*, 2019)).

# Materials and Methods

### scANVI: an extension to scVI for semi-supervised annotation

scVI is a hierarchical Bayesian model (Gelman & Hill, 2006) for single-cell RNA sequencing data with conditional distributions parametrized by neural networks. The graphical model of scVI (Fig 1C)

is designed to disentangle technical signal (i.e., library size discrepancies, batch effects) and biological signal. We propose in this manuscript an extension of the scVI model to include information about cell types in the generative model. We name this extension scANVI (single-cell ANnotation using Variational Inference).

**The generative model for scANVI**

In our generative model, we assume each cell $n$ is an independent realization of the following generative process. Let $K$ be the number of datasets and $C$ be the number of cell types across all datasets (including cell types that are not observed). Let **c** describe the expected proportion of cells for each cell type. As in general this information is not available to the user, we consistently use a non-informative prior c = $^1/C$ in the manuscript. Although some prior information about proportions of cell type is generally accessible, we observe that using the non-informative prior allows us to recover the correct proportion of cells. In addition, in comparative studies such as disease case-control comparisons, or between tissue comparisons of immune cells (Schafflick *et al*, 2020) we might not want to bias the estimate of cell type proportion by prior knowledge. All in all, adjustment of the prior **c** is not required. Latent variable.

$$c_n \sim \text{Multinomial}(\mathbf{c}) \tag{1}$$

describes the cell type of the cell *Normal*. Latent variable

$$u_n \sim \text{Normal}(0, I), \tag{2}$$

is a low-dimensional random vector describing cell $n$ within its cell type. Conceptually, this random variable could describe cell-cycles or sub-cell types. By combining cell type information $c_n$ and random vector $u_n$, we create a new low-dimensional vector

$$z_n \sim \text{Normal}(f_z^\mu(u_n, c_n), f_z^\sigma(u_n, c_n)), \tag{3}$$

where $f_z^\mu$ and $f_z^\sigma$ are two functions parametrized by neural networks. Let $s_n$ encode the dataset information. Given $l_\mu \in \mathbb{R}_+^K$ and $l_\nu \in \mathbb{R}_+^K$ specified per dataset as in (Lopez *et al*, 2018), latent variable.

$$l_n \sim \text{LogNormal}\left(l_\mu^{s_n}, l_\nu^{s_n}\right), \tag{4}$$

encodes a cell-specific scaling factor. As the prior are adjusted per dataset, our inference procedure will shrink the posteriors toward dataset-specific values. This is particularly useful when aligning datasets with dramatically different library size values. Let $\theta \in \mathbb{R}_+^G$ encode a gene specific inverse-dispersion parameter (inferred as in (Lopez *et al*, 2018)). Conditional distribution $x_{ng} \mid z_n, l_n, c_n, s_n$ is conform to the one from the scVI model

$$w_{ng} \sim \text{Gamma}\left(f_w^g(z_n, s_n), \theta_g\right) \tag{5}$$

$$y_{ng} \sim \text{Poisson}\left(l_n w_{ng}\right) \tag{6}$$

$$h_{ng} \sim \text{Bernoulli}\left(f_h^g(z_n, s_n)\right) \tag{7}$$

$$X_{ng} = \begin{cases} y_{ng} & \text{if } h_{ng} = 0 \\ 0 & \text{otherwise} \end{cases} \tag{8}$$

where $f_w$ and $f_h$ are functions parametrized by neural networks. $f_w$ has a final softmax layer to represent normalized expected frequencies of gene expression as in (Lopez *et al*, 2018). Let us note that the resulting distribution for the counts is zero-inflated negative binomial. However, it is straightforward using our implementation to use a negative binomial or a Poisson noise model instead. In this model, annotation $c_n$ can be either observed or unobserved following (Kingma *et al*, 2014; Louizos *et al*, 2016), which is useful in our applications where some datasets would come partially labeled or unlabeled. Only the first part of the generative model, as separated above, differs from the original scVI formulation. This corresponds to the top part of the new representation of the graphical model in Fig 1B.

**Approximate posterior inference for scANVI**

We rely on collapsed variational inference, a standard approximate Bayesian inference procedure that consists in analytically integrating over some of the random variables (Teh *et al*, 2007) before optimizing the parameters. As we proved in (Lopez *et al*, 2018), we can integrate the random variables $\{w_{ng}, y_{ng}, h_{ng}\}$ to simplify our model at the price of a looser though tractable lower bound ($x_{ng} \mid z_n, l_n, s_n$ is zero-inflated negative binomial). This procedure reduces the number of latent variable and avoids the need for estimating discrete random variables, which is a harder problem. We then use variational inference, neural networks and the stochastic gradients variational Bayes estimator (Kingma & Welling, 2014) to perform efficient approximate inference over the latent variable $\{z_n, u_n, c_n, l_n\}$. We assume our variational distribution factorizes as:

$$q_\Phi(c_n, z_n, l_n, u_n \mid x_n, s_n) = q_\Phi(z_n \mid x_n) q_\Phi(c_n \mid z_n) q_\Phi(l_n \mid x_n) q_\Phi(u_n \mid c_n, z_n). \tag{9}$$

Following (Kingma *et al*, 2014; Louizos *et al*, 2016), we derive two variational lower bounds: one L in the case of $c_n$ observed for $p_\Theta(x_n, c_n \mid s_n)$ and a second U in the case of $c_n$ nonobserved for $p_\Theta(x_n \mid s_n)$ where $\Theta$ are all the parameters (neural networks and inverse-dispersion parameters). Equations to derive the *evidence lower bound* (ELBO) are derived in Appendix Note E. We optimize the sum ELBO = L + U over the neural networks parameters and the inverse-dispersion parameters (in a variational Bayesian inference fashion). Remarkably, the approximate posterior $q_\Phi(c_n \mid z_n)$ can be used as a classifier, assigning cells to cell types based on the location on the latent space.

We sample from the variational posterior using the reparametrization trick (Kingma & Welling, 2014) as well as "minibatches" from the dataset to compute unbiased estimate of the objective gradients' with respect to the parameters. We use Adam (Kingma & Ba, 2015) as a first-order stochastic optimizer to update the model parameters.

**Choice of hyperparameters**

For all harmonization tasks in this paper, we consistently use the same set of hyperparameters. Each network has exactly 2 fully connected layers, with 128 nodes each. The number of latent dimensions is 10, the same as other algorithms for benchmarking

purposes (e.g., the number of canonical correlation vectors used in Seurat Alignment). The activation functions between two hidden layers are all ReLU. We use a standard link function to parametrize the distribution parameters (exponential, logarithmic or softmax). Weights for the first hidden layer are shared between $f_w$ and $f_h$. We use Adam with $\eta = 0.001$ and $\epsilon = 01$. We use deterministic warmup (Sønderby *et al*, 2016) and batch normalization (Ioffe & Szegedy, 2015) in order to learn an expressive model. When we train scANVI, we therefore assume that the data come from a set of $C_{observed} + C_{unobserved}$ populations, each generated by a different distribution of $z_n$ values. This set includes the $C_{observed}$ populations for which annotated cells are available, and $C_{unobserved}$ population that accounts for cell types for which an annotation is not available to the algorithm. Ad hoc training procedures for scANVI inference are described in Appendix Note B.

## Hierarchical classification of cells onto a cell type taxonomy

For hierarchical label propagation in scANVI, we propose an extension of the formerly presented model by modifying the variable $c_n$ to be a tuple where each entry denotes the label at a given level of the hierarchy. Our approach is similar to previous work in robustness to noisy labels (Goldberger & Ben-Reuven, 2017) and hierarchical multi-labels flavors of classification problems (Wehrmann *et al*, 2018). We detail the case for a depth of level two in Appendix Note D though our approach can in principle be adapted to arbitrary depths.

## Bayesian differential expression

### Extending differential expression for scVI to the case of multiple batches

For each gene $g$ and pair of cells $(z_a, z_b)$ with observed gene expression $(x_a, x_b)$ and dataset identifier $(s_a, s_b)$, we can formulate two mutually exclusive hypotheses:

$$\mathcal{H}_1^g := E_s f_w^g(z_a, s) > E_s f_w^g(z_b, s) \text{ vs. } \mathcal{H}_2^g := E_s f_w^g(z_a, s) \leq E_s f_w^g(z_b, s), \tag{10}$$

where the expectation $E_s$ is taken with the empirical frequencies. Notably, we propose a hypothesis testing that do not calibrate the data to one batch but will find genes that are consistently differentially expressed. Evaluating which hypothesis is more probable amounts to evaluating a Bayes factor (Held & Ott, 2018) (Bayesian generalization of the p-value) which is expressed as:

$$K = \log_e \frac{p(\mathcal{H}_1^g | x_a, x_b)}{p(\mathcal{H}_2^g | x_a, x_b)}. \tag{11}$$

The sign of $K$ indicates which of $\mathcal{H}_1^g$ and $\mathcal{H}_2^g$ is more likely. Its magnitude is a significance level and throughout the paper, we consider a Bayes factor as strong evidence in favor of a hypothesis if $|K| > 3$ (Kass & Raftery, 1995) (equivalent to an odds ratio of $exp$ (3) $\approx 20$). Notably, each of the probabilities in the likelihood ratio for $K$ can be written as:

$$p(\mathcal{H}_1^g | x_a, x_b) = \sum_s \int \int_{(z_a, z_b)} 1_{f_w^g(z_a, s) \leq f_w^g(z_b, s)} p(s) dp(z_a | x_a) dp(z_b | x_b), \tag{12}$$

where $p(s)$ designated the relative abundance of cells in batch $s$ and all of the measures are low dimensional. Since we cannot in principle achieve efficient posterior sampling, the naive Monte Carlo estimator obtained by replacing the real posterior $p(z \mid x)$ by the variational posterior $q_\Phi(z \mid x)$ is biased. The resulting Bayes factors are therefore approximate though yield very competitive performance, as explained in the original publication of scVI (Lopez *et al*, 2018). Since we assume that the cells are independently distributed, we can average the probabilities for the hypotheses across a large set of randomly sampled cell pairs, one from each subpopulation. The Bayes factor from the averaged probability will provide an estimate of whether cells from one subpopulation tend to express $g$ at a higher frequency.

### Differential expression with scANVI

In the case of scANVI, we need not rely on specific cells since labels are given during the training. We still use the generative model but with the following probability for $p(\mathcal{H}_1^g | c_a, c_b)$ where $c_a$ (resp. $c_b$) is the first (resp. second) cell type of interest:

$$p(\mathcal{H}_1^g | c_a, c_b) = \sum_s \int 1_{f_w^g(z_a, s) \leq f_w^g(z_b, s)} p(s) dp(z_a | u_a, c_a) \\ dp(z_b | u_b, c_b) dp(u_a) dp(u_b). \tag{13}$$

Notably, we draw here data from the prior distribution and not the posterior for given cells. As a consequence, these Bayes factors can be approximated in a unbiased fashion using a naive Monte Carlo estimator. We noticed in the case of the real dataset that the aggregate posterior on $u$ might not perfectly match the prior for rare cell types. Consequently, we replaced the prior by the aggregate posterior for all the analyses in this manuscript.

## Datasets

We report an extensive list of datasets at Appendix Table S1. For all UMI based datasets we took the raw counts without any normalization as input to scVI.

### Gene selection

A common practice in data harmonization is to perform gene selection prior to harmonization. This assumption is critical when the number of genes that can be taken into account by the algorithm is small and potentially biological signal could be lost. scVI is however designed for large datasets which do not fall into the high-dimensional statistics data regime (Lopez *et al*, 2018). Remarkably, there is no need for crude gene filtering as part of our pipeline and we adopt it as part of this publication only for concerns of fairness in benchmarking. For real datasets, we calculated the dispersion (variance to mean ratio) for all genes using Seurat in each dataset and selected $g = 1,000$ genes with the highest dispersion from each. The performance of scVI is not as affected by gene set and we use the same gene selection scheme as in (Butler *et al*, 2018) to ensure fairness in our comparison. We then took the union of these gene list as input to Seurat Alignment, MNN and scANVI. One exception is the differential expression study for which we kept the gene set ($g = 3,346$) to have it match the bulk reference as in (Lopez *et al*, 2018).

### Cell type labeling for the Tabula Muris Dataset

For the Tabula Muris dataset, cell types are defined by first reducing the dimensions of the data by principal component analysis and then performing nearest-neighbor-graph-based clustering. The labels for Smart-Seq2 and 10x data are derived independently. All cells in both dataset are labeled, but there is also a possibility that they are mislabelled since the labels are computationally derived. Since cells used in Smart-Seq2 are first FACS sorted into each plate, some cell types might have been lost during the sorting process, resulting in incomplete overlap in cell types between the two datasets.

### Hierarchical cell type labeling for the mouse nervous system dataset

The multi-level labels are generated through an iterative process that is described in detail in the original publication (Zeisel *et al*, 2018). The clustering was performed with strict quality filters, takes into account anatomical information and were validated at different levels using existing scRNA-seq dataset, osmFISH, RNAscope and others. The cell types taxonomy is derived differently for each level and the details can be found in the original publication. Cell type clusters were obtained by Louvain clustering on a multiscale $k$-nearest neighbors graph and DBSCAN. The first level separates neurons and non-neuronal cells. The second level separates peripheral neuronal system from central neuronal system. The third layer separates anterior posterior domain, and the fourth layer is split by excitatory versus inhibitory neurotransmitter. At this level, all cells are divided into 39 subsets, each corresponding to a coarse cell type definition. Then, within each subset the authors defined ($N = 28$) enriched genes and used linkage (correlation distance and Ward method) to construct the dendrogram.

### Normalization of CITE-seq data

Since we did not explicitly model the CITE-seq data in scVI or scANVI, we normalized it by fitting a Gaussian mixture model to each individual protein with two components. We then transformed each individual protein count as $x \mapsto (x - (\mu_1 + \mu_2/2))_+$ where $\mu_1$ and $\mu_2$ designate the mean of the mixtures and $._+$ is the positive part of a real number.

### Normalization of SmartSeq2 data

For the MarrowMT-ss2 dataset, we normalized the read counts per gene by relative transcript length (average transcript lengths of a gene divided by average gene length over all genes), and subsequently took the integer part of the normalized count. This is different from standard normalization procedures in that we do not normalize by cell size because cell size normalization can be performed by scVI. And we only keep the integer part of the counts, due to the distributional assumptions made by scVI. The scVI model can to be extended to fit data with amplification bias, however we have not done so for this paper and thus have to perform this normalization heuristic.

### Simulation of continuous gene expression using SymSim

First we simulated the true expression matrix for a tree with 5 cell types using the function SimulateTrueCounts. Instead of sampling cells only from the leaf populations, we uniformly sample cells along all branches by using the parameter evf type="continuous".

We then added noise to the data with the function True2Observed-Counts with the parameters:
protocol="nonUMI", alpha_mean = 0.1, alpha_sd = 0.05, rate_2PCR = 0.7, nPCR1 = 16,depth_mean = 1e5, depth_sd = 3e3.

### Simulation for DE benchmark using SymSim

First we simulated the true expression matrix for 20,000 cells from 5 cell types using the function SimulateTrueCounts. We then randomly split the cells into two batches. We then added noise to the data the function True2ObservedCounts with the parameters:

Batch 1: protocol="UMI", alpha_mean = 0.03, alpha_sd = 0.009, gene_len = gene_len, depth_mean = 5e5, depth_sd = 1.5e4.

Batch 2: protocol="UMI", alpha_mean = 0.1, alpha_sd = 0.03, gene_len = gene_len, depth_mean = 1e6, depth_sd = 1.5e5.

## Algorithms for benchmarking

### Seurat Alignment

We applied the Seurat Alignment procedure from the R package Seurat V2. The number of canonical correlation vectors is 10 for all the datasets, which is also identical to the number of latent dimensions used for scVI and scANVI.

### Seurat PCA

We applied the Seurat PCA procedure from the R package Seurat V2. This method is a simple PCA based after normalization by Seurat. Seurat PCA is used to obtain the individual dataset latent space to evaluate the $k$-nearest neighbors purity for all non-scVI based methods. The number of principal components is 10.

### Matching Mutual Nearest Neighbors

We used the mnnCorrect function from https: //www.rdrr.io/bioc/ scran/man/mnnCorrect.html with default parameters. In order to compare with other methods, we applied a PCA with 10 principal components on the output of the batch-corrected gene expression matrix.

### scmap

We applied the scmap-cluster procedure from the R package scmap. As the scmap manuscript insists heavily on why the M3Drop (Andrews & Hemberg, 2019) gene filtering procedure is crucial to overcome batch effects and yield accurate mapping, we let scmap choose its default number of genes ($g = 500$) with this method.

### ComBat

We used the R package sva with default parameters.

### UMAP

We used the umap class from the UMAP package with a default parameters and spread = 2.

### DBSCAN

We used the DBSCAN algorithm from the Python package from the python package scikit-learn V0.19.1 and we searched for an optimal hyperparameter combination by a grid search over eps and min_samples from the range of $0.1 - 2$ and $5 - 100$ respectively.

Although some combinations of parameters yield more than one clusters, the smaller clusters comprise of less than 1% of the data. We then evaluated DBSCAN with eps = 1.23, min_samples = 10 and default values for all other hyperparameters.

### PhenoGraph

We used the phenograph.cluster function from the Python package PhenoGraph 1.5.2 downloaded from https://github.com/jacoble vine/PhenoGraph with its default parameters.

### CORAL

We used the implementation from https://github.com/jindongwa ng/transferlearning/tree/master/code/traditional/CORAL.

### MAGAN

We used the implementation from https://github.com/Krishnaswa myLab/MAGAN.

### Harmony

We used the implementation from https://github.com/immunoge nomics/harmony.

### Scanorama

We used the implementation from https://github.com/brianhie/sca norama.

## Evaluations metrics

### Entropy of batch mixing

Fix a similarity matrix for the cells and take $U$ to be a uniform random variable on the population of cells. Take $B_U$ the empirical frequencies for the 50 nearest neighbors of cell $U$ being a in batch $b$. Report the entropy of this categorical variable and average over $T = 100$ values of $U$.

### k-nearest neighbors purity

Compute two similarity matrices for cells from the first batch, one from the latent space obtained with only cells from the first batch and the other from the latent space obtained using both batches of cells. We always rely on the Euclidean distance on the latent space. Take the average ratio of the intersection of the $k$-nearest neighbors graph from each similarity matrix over their union. Compute the same statistic for cells from the other batch and report the average of the two.

### Weighted and unweighted accuracy

We evaluate the accuracy of cell type classification algorithms by comparing the predictions to previously published labels. The unweighted accuracy is the percentage of cells that have the correct label. The weighted accuracy corresponds to first calculating accuracy for each cell type, and then averaging it across cell types. The weighted accuracy assigns the same weight to each cell type and thus weighs correct prediction of rare cell types more heavily than the unweighted accuracy. We report the weighted accuracy throughout this manuscript.

### Maximum Posterior Probability

We evaluate the performance of the scANVI classifier at transferring labels from an annotated dataset to an unannotated

dataset by looking at the maximum posterior probability for the observed classes. By default scANVI classifier sets the number of classes to the same number of cell types in the merged dataset. In the case of $N$ observed labels from the annotated dataset and one unannotated dataset (thus the cell type label is "Unlabeled") scANVI assumes $N + 1$ classes. For each cell, scANVI assigns a posterior probability for each of the $N + 1$ classes. The maximum posterior probability for the observed classes is the highest probability of a cell being assigned to one of the $N$ observed classes.

## Signature for sub-division of T cells in human PBMCs

### Gene sets

For ranking the cells, we used both positive and negative sets of genes:

- CD4 Regulatory: $GITR^+$ $CTLA4^+$ $FOXP3^+$ $CD25^+$ $S100A4^-$ $CD45^-$ $CD8B^-$
- CD4 Naive: $CCR7^+$ $CD4^+$ $S100A4^-$ $CD45^-$ $FOXP3^-$ $IL2RA^-$ $CD69^-$
- CD4 Memory: $S100A4^+$ $CD25^-$ $FOXP3^-$ $GITR^-$ $CCR7^-$
- CD8 Naive: $CD8B^+$ $CCR7^+$ $CD4^-$

### Signature calculus

To compute the signature of a cell, we followed the normalization procedure from (DeTomaso & Yosef, 2016) which consists in dividing by total numbers of UMIs, applying a entry-wise transformation $x \rightarrow \log(1 + 10^4 x)$ and $z$-score normalization for each gene. Then, we aggregated over the genes of interest for each cell by applying the sign from the gene set and averaging.

# Data availability

All of the datasets analyzed in this manuscript are public and referenced at https://github.com/chenlingantelope/Harmonization SCANVI.

In addition, an open-source software implementation of scVI and scANVI is available on GitHub in the new scvi-tools repository (https://github.com/YosefLab/scvi-tools). All code for reproducing results and figures in this manuscript is deposited at: https://doi. org/10.5281/zenodo.2529945.

**Expanded View** for this article is available online.

## Acknowledgments

CX, RL, and NY were supported by grant U19 AI090023 from NIH–NIAID and U19 MH114821 NIMH. We thank Maxime Langevin, Yining Liu and Jules Sama-ran for helpful discussions and early work on the scVI codebase as well as Allon Wagner and Chao Wang for their help on the choice for high specificity genes in the T cell study. Additionally, we would like to thank Adam Gayoso and Galen Xing for discussions around the stability of the algorithm as well as the API within the new scvi-tools package.

## Author contributions

RL, EM, JR and NY conceived the statistical model. EM developed the software. CX, RL and EM applied the software to real data analysis. CX, RL, JR, NY, and MIJ wrote the manuscript. NY and MIJ supervised the work.

## Conflict of Interest

The authors declare no competing interests.

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
