## [Review Process File · Molecular Systems Biology]

Harmonization and Annotation of Single-cell Transcriptomics Data with Deep Generative Models

Chenling Xu, Romain Lopez, Edouard Mehlman, Jeffrey Regier, Michael Jordan, and Nir Yosef
DOI: [10.15252/msb.20209620](https://doi.org/10.15252/msb.20209620)

Corresponding author(s): Nir Yosef (niryosef@berkeley.edu)

Review Timeline:

Submission Date:	8th Apr 20
Editorial Decision:	6th Jun 20
Revision Received:	30th Sep 20
Editorial Decision:	9th Nov 20
Revision Received:	18th Nov 20
Accepted:	26th Nov 20

Editor: Jingyi Hou

Transaction Report:

Thank you for submitting your work to Molecular Systems Biology. We have now heard back from two of the three reviewers who agreed to evaluate your manuscript. Unfortunately, after a series of reminders we did not manage to obtain a report from Reviewer #3. In the interest of time, and since the recommendations of the other two reviewers are quite similar, I prefer to make a decision now rather than further delaying the process. As you will see below, the reviewers acknowledge the potential interest of the presented methodology. They raise however a series of concerns, which we would ask you to address in a major revision.

I think that the reviewers' recommendations are rather clear and there is therefore no need to reiterate the comments listed below. In light of the concerns of Reviewer # 2, we would ask you to edit the manuscript to make sure that the main findings are sufficiently clear and easily accessible to the general audience of Molecular Systems Biology. To further improve the accessibility of the presented method to the community, we would strongly encourage you to include a tutorial as recommended by Reviewer #1.

All other issues raised by the reviewers need to be satisfactorily addressed as well. As you may already know, our editorial policy allows in principle a single round of major revision and it is therefore essential to provide responses to the reviewers' comments that are as complete as possible.

On a more editorial level, please do the following.

REFeree REPORTS

Reviewer #1:

In this manuscript the authors successfully tackle several of the most important challenges in high-throughput single-cell transcriptomic studies within a unified framework. Specifically, the authors (1) provide extensive analysis to show that their previously developed probabilistic framework works well for the integration of data across datasets to enable e.g. joint clustering, (2) provide a novel framework to generate automatic assignment of labels to cells within (merged) datasets via semi-supervised learning, and (3) show how the learned probabilities from their introduced frameworks can be used to perform probabilistic decision tasks within a uniform space across datasets.

In this work the authors explain their introduced framework, its extension and the performance across the tasks detailed above in great depth and detail. The authors work is thorough with respect to both model design and theoretical basis as well as experimental testing. Importantly, the authors show favorable performance on multiple different data set types, and show favorable performance on those datasets in a variety of tasks, comprehensively demonstrating the power of their approach. I believe that this framework will be broadly used by the single-cell transcriptomic

community for both broad characterization tasks e.g. within the HCA effort, as well as by specific studies tailored to study specific biological hypotheses. Below are comments that I believe will improve this manuscript before publication.

Major:

There are recently been publications discussing the possibility that models that do not incorporate zero inflation are suitable in the analysis of 10X single-cell RNA-seq data. E.g.

"Normalization and variance stabilization of single-cell RNA-seq data using regularized negative binomial regression", Hafemeister and Satija

"Feature selection and dimension reduction for single-cell RNA-Seq based on a multinomial model" Townes et al

The authors should mention that their model does assume zero inflation, but cite/relate to these works.

The authors describe the method and framework in great detail, but several components still require better explanations. For example, in supplementary note 3, more details are needed to understand the approach that the authors took in algorithm 2. How are the iterations occurring (outer loop, waiting for both elements to converge) if each of the components have converged in the inner loops?

Additionally, there is a typo in the gradient of both algorithms in this note.

In the methods section, the authors note the scVI and scANVI could be used on all genes, but that the authors took the took 1K genes to be comparable to other methods (in most analysis). I would like to know if analyzing with all genes (or a considerably larger number of genes) (1) is feasible to run (doesn't require too long to compute or too many resources) and (2) if performance is better when ran on all genes.

The importance and usability of the authors model and method are not dependent on this point, but because the authors elude to the possibility that running their methods with all genes might be favorable, it would be good for the readers/users to know what would be preferable to use/try on their data.

Supplementary Table 2 does not have measurements for all 3 metrics to compare performance, but only for one metric (and it's not clear which one).

Also, what is scVI_nb in that table? I assume it's negative binomial, but I do not believe this was mentioned in the text?

For comparisons such as in Figure 2 (c) a helpful way to provide "head-to-head" comparisons would be to pick e.g. 3 values of K, and for each of these values plot, for each method, the Entropy of batch matching vs. kNN purity. This could give readers/users a way of assessing which methods are dominantly better on both metrics, and which might be stronger on one metric and compromise the other (this analysis has in mind frameworks in which there is a tradeoff, such as plotting sensitivity/specificity measures). To do this analysis one would need to fix K, but with three reasonable choices of K we could have a nice visual comparing the methods.

The results discussed in the final part of the results section, which discusses conducting differential expression using the latent space, should be represented in the main figures associated with the text. The analyses presented by the authors is important and informative, and the results should be highlighted as part of the main figures.

This manuscript introduces a method for use by the community. As such, this reviewer believes that the method accompanied by a detailed tutorial (doing a walk-through of an analysis of a published dataset) should be available to be reviewed before the manuscript is accepted.

Minor:

There are several typos throughout the paper, as well as grammatical errors.

The methods section should be better organized to have methods in the order in which they are referenced in the main text.

Improve the explanation of the following sections:

Entropy of batch mixing. The procedure wasn't clear to me from the text.

k-nearest neighbors purity

Page 7, end of first paragraph. Please spell out what you mean / what to observe in the comparison.

Reviewer #2:

In this paper, Xu et al present scVI, as an effective method for harmonization and integration of single cell data from different modalities. They show that it compares favorably to currently available methods. Further, they describe scANVI, a semi-supervised variant of scVI, that has the capability of utilizing prior cell state/label information to solve the annotation problem in single cell RNAseq data integration. The benchmarking done by the authors to provide support for their method is extensive and compelling, and shows that both scVI and scANVI have superior performance in data integration based on KNN purity and entropy of batch mixing but also has the ability to scale to accommodate datasets with a large number of cells. The manuscript is very well organized and extensively described with 21 supplementary figures and six supplementary notes. I believe that with the few changes mentioned below will be a great contribution to the literature and therefore a fitting publication in Molecular Systems Biology.

The authors do not provide a good enough intuition for how scVI actually works.

How do the two latent variables actually vary. Can Figure 1 be improved to provide better insight into the logic of method? Furthermore, to allow for a wider reach of the paper, i.e to computational biologists/mathematicians as well as biologists working with single cell data, it would be useful to modify the language of the manuscript to be more accessible to broader audiences. Perhaps the text can be improved by reducing the usage of jargon and use of shorter, non complex sentences to convey the meaning in a more intuitive manner to non-experts. Detailed methods can be included in the supplementary to provide information as required.

In Figure 2, the authors show that the method outperforms Seurat Combar and MNN. However, in the remaining analyses they only compare to Seurat. Since there are many other methods - such as Harmony, limma, scGen, Scanorama, MMD-ResNet, ZINB-WaVE, scMerge, LIGER, and BBKNN - as described in Tran et al Genome Biology 2020 - the authors should further support the notion that their approach can really offer an advantage to these.

The examples used by the authors to evaluate the performance of their tool are apt and show that the tool performs well in scenarios where cell types are fairly well distinguished. However, the question remains regarding how the tool performs in cases the cell types are not clearly demarcated. To this end, it would be useful to evaluate the performance of the tool on additional datasets that are intrinsically more heterogenous and data quality is typically poor, e.g. tumor dataset or alternatively, on a simulated dataset with additional noise or reduced data quality to evaluate the robustness of scVI and scANVI in such a scenario.

The strength of this paper is that they demonstrate their approach on many different datasets. Apart from the use cases provided in the manuscript by the authors, it would be useful to see if the tool can be further extended for the integration of data from different species. For example, a tool that allows for the integrative study of scRNAseq data from human and mouse. This would be of high significance and such a tool would be a valuable resource of researchers working with model organisms.

A few typos found in the document:

1. "While we demonstrate that scVI performs well in these scenarios, we also demonstrate that the latent space learned by scANVI provides a proper harmonized representation of the input datasets -" Do they mean "learned"
2. In Figure 2, the word "trajectory" is misspelled in 2 places.

Reviewer Comments:

Reviewer #1:

In this manuscript the authors successfully tackle several of the most important challenges in high-throughput single-cell transcriptomic studies within a unified framework. Specifically, the authors (1) provide extensive analysis to show that their previously developed probabilistic framework works well for the integration of data across datasets to enable e.g. joint clustering, (2) provide a novel framework to generate automatic assignment of labels to cells within (merged) datasets via semi-supervised learning, and (3) show how the learned probabilities from their introduced frameworks can be used to perform probabilistic decision tasks within a uniform space across datasets.

In this work the authors explain their introduced framework, its extension and the performance across the tasks detailed above in great depth and detail. The authors work is thorough with

respect to both model design and theoretical basis as well as experimental testing. Importantly, the authors show favorable performance on multiple different data set types, and show favorable performance on those datasets in a variety of tasks, comprehensively demonstrating the power of their approach. I believe that this framework will be broadly used by the single-cell transcriptomic community for both broad characterization tasks e.g. within the HCA effort, as well as by specific studies tailored to study specific biological hypotheses. Below are comments that I believe will improve this manuscript before publication.

Major:

There have recently been publications discussing the possibility that models that do not incorporate zero inflation are suitable in the analysis of 10X single-cell RNA-seq data. E.g.

"Normalization and variance stabilization of single-cell RNA-seq data using regularized negative binomial regression", Hafemeister and Satija "Feature selection and dimension reduction for single-cell RNA-Seq based on a multinomial model" Townes et al. The authors should mention that their model does assume zero inflation, but cite/relate to these works.

In the submitted version, we mentioned zero-inflation in the **5th paragraph of the discussion** (page 13) and in the **Appendix**. Briefly, we showed in **Appendix Note C** and **Appendix Figure 20 and 21** that in most cases Zero Inflated Negative Binomial (ZINB) does perform similarly to NB, with the exception of Smart-Seq2 data. In the revised version, we added a summary of the findings of **Appendix Note C** to the discussion section. In this summary, we also cited those two papers and say that our investigation is consistent with their conclusion.

The authors describe the method and framework in great detail, but several components still require better explanations. For example, in supplementary note 3, more details are needed to understand the approach that the authors took in algorithm 2. How are the iterations occurring (outer loop, waiting for both elements to converge) if each of the components have converged in the inner loops?

Additionally, there is a typo in the gradient of both algorithms in this note.

We have arranged **Appendix Note B** to better explain how the training procedure works. In particular, we use a fixed number of iterations that ensures convergence in practice. This could be further improved via early stopping criterion. While this is beyond the scope of this manuscript, we implemented this option in the new software release of scANVI. Regarding the typo, we have deleted the algorithm box and instead added descriptions to the main body of the text. We have found that this increased clarity.

In the methods section, the authors note the scVI and scANVI could be used on all genes, but that the authors took the took 1K genes to be comparable to other methods (in most analysis). I would like to know if analyzing with all genes (or a considerably larger number of genes) (1) is feasible to run (doesn't require too long to compute or too many resources) and (2) if performance is better when ran on all genes.

The importance and usability of the authors model and method are not dependent on this point, but because the authors elude to the possibility that running their methods with all genes might be favorable, it would be good for the readers/users to know what would be preferable to use/try on their data.

While scVI and scANVI both accommodate for large gene sets in terms of run time, we usually recommend filtering genes for best performance, especially when the dataset has a low number of cells. As a rule of thumb, performance starts to decrease when the number of genes becomes comparable or lower than the number of cells. Notably, this point is discussed in detail in a recent comparative analysis (not from our group) of data integration algorithms for scRNA-seq data [Luecken et al., 2020]. We have now added a reference to this new preprint. To address the concern on runtime, we ran the algorithms used in this paper with 500 up to 8000 genes. The results of the run time is presented in **Appendix Table 4**. With these runs, we find that the number of genes has only a mild (sub-linear) effect on the run time.

[Luecken et al., 2020] Luecken, Malte D., et al. "Benchmarking atlas-level data integration in single-cell genomics." *BioRxiv* (2020). <https://doi.org/10.1101/2020.05.22.111161>

Supplementary Table 2 does not have measurements for all 3 metrics to compare performance, but only for one metric (and it's not clear which one). Also, what is scVI_nb in that table? I assume it's negative binomial, but I do not believe this was mentioned in the text?

Indeed, **Appendix Table 2** did not contain the three metrics but only one metric, which we considered to be a third alternative for the two metrics that are displayed in Figure 2. As a reminder, this metric quantified for the retainment of structure post-harmonization, in which the k-nearest neighbor overlap is replaced by the overlap of a k-means clustering. We have now restructured **Figure 2** to include this metric so that it is presented in parallel with the other metrics. We therefore excluded the former Appendix Table 2 from the current submission.

About scVI_nb, it was indeed a follow-up analysis to show that the results from the negative binomial version (NB) of scVI (vs. zero inflated NB, which is default in this paper) are unchanged. Even though we removed the appendix table above, the NB performance included in **Appendix Figure 20 and 21** is sufficiently supporting our claim in **Appendix Note C**.

For comparisons such as in Figure 2 (c) a helpful way to provide "head-to-head" comparisons would be to pick e.g. 3 values of K, and for each of these values plot, for each method, the Entropy of batch matching vs. kNN purity. This could give readers/users a way of assessing which methods are dominantly better on both metrics, and which might be stronger on one metric and compromise the other (this analysis has in mind frameworks in which there is a tradeoff, such as plotting sensitivity/specificity measures). To do this analysis one would need to fix K, but with three reasonable choices of K we could have a nice visual comparing the methods.

We incorporated the figures mentioned by the reviewer for a fixed value of the knn purity for k=150 (as well as the k-means preservation, as a third metric). These are now added in **Figure**

2. When choosing multiple values of K the comparative performance between different algorithms remain similar, therefore we only show one value of K in the main figure.

Also, we removed the UMAP panels from the main figure. These are now all delayed in **Appendix Figures 4, 5, 6, 7**.

The results discussed in the final part of the results section, which discusses conducting differential expression using the latent space, should be represented in the main figures associated with the text. The analyses presented by the authors is important and informative, and the results should be highlighted as part of the main figures.

To address this comment, we have moved the former **Appendix Figure 17** to the main text (**Figure 7**).

This manuscript introduces a method for use by the community. As such, this reviewer believes that the method accompanied by a detailed tutorial (doing a walk-through of an analysis of a published dataset) should be available to be reviewed before the manuscript is accepted.

Since the release of this manuscript on Biorxiv, we have spent a considerable amount of time on our codebase. We have very recently announced a new version of the package (scvi-tools V1.0). scVI and scANVI are now implemented in scvi-tools (<https://scvi-tools.org>), which has an improved interface, tutorials, and integration with the widely-used Scanpy package. As a result, scVI and scANVI will be more easily accessible to R users, as the interoperability of data objects has improved. It will also be easier for use by Python users, as we added a seamless integration of all our packages with AnnData and the scanpy ecosystem and remove the requirement for the PyTorch deep learning library directly.

We host all our tutorials on a GitHub submodule [1]. A first tutorial of interest is a walkthrough of scVI and scANVI for the Tabula Sapiens Bone Marrow dataset (runnable in Colab [2]). A second tutorial is the seed labelling example for annotation of T cells described in this manuscript (runnable in Colab [3]). We have also incorporated these tutorials into a user guide (work in progress) [4].

[1] <https://github.com/YosefLab/scvi-tutorials>

[2] <https://colab.research.google.com/github/yoseflab/scvi-tutorials/blob/master/harmonization.ipynb>

[3] https://colab.research.google.com/github/yoseflab/scvi-tutorials/blob/master/seed_labeling.ipynb

[4] https://www.scvi-tools.org/en/latest/user_guide/index.html

Minor:

There are several typos throughout the paper, as well as grammatical errors.

We have identified several grammatical errors in the abstract, the figure legends, as well as minor typos throughout the manuscript. We have corrected them.

The methods section should be better organized to have methods in the order in which they are referenced in the main text.

We renamed the subsection in the methods section and moved things to match the presentation of the main text. First, we present scANVI and its components in order of appearance in the manuscript (model, inference, hyperparameters, hierarchical labels, and then differential expression). Second, we present the datasets, with details given in the order of appearance in the manuscript. Third, we present the remaining topic in order of appearance.

Improve the explanation of the following sections:

Entropy of batch mixing. The procedure wasn't clear to me from the text.

k-nearest neighbors purity

Page 7, end of first paragraph. Please spell out what you mean / what to observe in the comparison.

We added more details in the first paragraph on **page 7** to address the two previous comments, both on how the three measurements are computed, the range of the measurements and which aspect of the performance they measure.

We also deleted the UMAP plots in **Figure 2** in order to make room for figures that can clearly show the trade-off between batch mixing and preservation of biological variation.

Reviewer #2:

In this paper, Xu et al present scVI, as an effective method for harmonization and integration of single cell data from different modalities. They show that it compares favorably to currently available methods. Further, they describe scANVI, a semi-supervised variant of scVI, that has the capability of utilizing prior cell state/label information to solve the annotation problem in single cell RNAseq data integration. The benchmarking done by the authors to provide support for their method is extensive and compelling, and shows that both scVI and scANVI have superior performance in data integration based on KNN purity and entropy of batch mixing but also has the ability to scale to accommodate datasets with a large number of cells. The manuscript is very well organized and extensively described with 21 supplementary figures and six supplementary notes. I believe that with the few changes mentioned below will be a great contribution to the literature and therefore a fitting publication in Molecular Systems Biology.

The authors do not provide a good enough intuition for how scVI actually works.

How do the two latent variables actually vary. Can Figure 1 be improved to provide better insight into the logic of method? Furthermore, to allow for a wider reach of the paper, i.e to computational biologists/mathematicians as well as biologists working with single cell data, it would be useful to modify the language of the manuscript to be more accessible to broader audiences. Perhaps the

test can be improved by reducing the usage of jargon and use of shorter, non complex sentences to convey the meaning in a more intuitive manner to non-experts. Detailed methods can be included in the supplementary to provide information as required.

We addressed this concern in three different ways. First, we have edited the presentation of scVI and scANVI in order to better explain the relationship between the most important latent variables (joint modeling of scRNA-seq datasets section). Second, we have edited **Figure 1** to better reflect the nature of the two tools that we describe in this manuscript, as well as what are the different use cases for them. Third, we have detected and removed machine learning jargon from the main text. This has been either moved to the methods section or to the supplementary notes.

In particular, we have identified and edited the following sections. (A) in the paragraph of the introduction section where we discuss other harmonization methods. We have now delayed the technical presentation to the supplements. In particular, we restructured **Appendix Notes A** into a unique related work section. In the main text, we solely focus on explaining conceptually why this is a hard problem and why current approaches have drawbacks (B) in the paragraph of the introduction section where we present scANVI, we introduce the principle of semi-supervised learning before explicitly mentioning it. (C) In the first paragraph of the section “joint modeling of scRNA-seq datasets”, we reduce the mathematical presentation. We instead focus on the input to the scANVI model (gene expression, batch identifier, partial cell type information). (D) in the 6th paragraph of the discussion section, we removed the conversation about interpretability, which was overly technical. Instead, we focus on the importance of quantifying uncertainty in multi-omics data integration.

In Figure 2, the authors show that the method outperforms Seurat Combar and MNN. However, in the remaining analyses they only compare to Seurat. Since there are many other methods - such as Harmony, limma, scGen, Scanorama, MMD-ResNet, ZINB-WaVE, scMerge, LIGER, and BBKNN - as described in Tran et al Genome Biology 2020 - the authors should further support the notion that their approach can really offer an advantage to these.

To address this comment, we have now added to our harmonization benchmark (Figure 2, using four different test cases) two of the most prevalent methods - namely Harmony and Scanorama. Overall, we observed good performance from scVI and scANVI compared to these methods, providing a tradeoff between batch correction and retainment of original structure (see new **Figure 2** and **Appendix Figure 4, 5, 6, 7**).

Notably, an additional benchmark paper that recently became available (Luecken et al.) proposed diverse set of benchmarking regimes for the harmonization task, finding scVI to perform well, compared to a large cohort of methods, including some of the ones specified above.

The examples used by the authors to evaluate the performance of their tool are apt and show that the tool performs well in scenarios where cell types are fairly well distinguished. However, the question remains regarding how the tool performs in cases the cell types are not clearly

demarcated. To this end, It would be useful to evaluate the performance of the tool on additional datasets that are intrinsically more heterogenous and data quality is typically poor, e.g. tumor dataset or alternatively, on a simulated dataset with additional noise or reduced data quality to evaluate the robustness of scVI and scANVI in such a scenario.

We agree with the reviewer that the scenarios where cell types are not clearly demarcated are the more challenging (and computationally interesting!) applications. We believe that the paper already includes a number of tests that pertain to the reviewer's concern, namely - cases in which the cells are not clearly stratified into groups, as well as a test for robustness to errors in a-priori labeling:

1. For the harmonization task, the manuscript introduces performance metrics that are agnostic for any a-priori stratification of cells (e.g., into clusters or sub-populations). Instead, we employ local measures that quantify: (i) the average mixing between batches amongst all local environments (KNN of each cell, for different values of K); and (ii) the retainment of original structure in local neighborhoods (KNN purity). See **Figure 2**.
2. The manuscript includes several evaluations of the algorithms on instances in which the cells are not stratified into distinct, well separated groups:
 - a. Harmonization of continuous trajectories using developmental dataset (**Figure 5**). Here, the cells are not clearly stratified into distinct clusters, but rather form continuous gradients. Nevertheless, we see superior performance of scVI and scANVI in both performance metrics summarized in #1 above.
 - b. We perform benchmark over the task of cell state annotation in continuous trajectories using simulated datasets (**Appendix Figure S17**).
 - c. Annotation of T cell subsets (**Figure 6**). In this test, the populations of T cells have been identified experimentally, however, in transcriptome space, the different subtypes are not clearly separable. Here, we demonstrate superior performance of scANVI in the context of expanding seed labeling (i.e., we are able to predict T cell subtypes based on a few seed labels generated with well known markers).
 - d. Related to that, we also tested the robustness of our differential expression analysis to errors in the seed labeling, using label-corruption in both simulated and real data (**Appendix Figures S21**).

As response to this comment, we repeated the benchmark of harmonization of continuous trajectories using developmental dataset (see 2a above), while subsampling only a subset of the UMIs. We added a new simulation to test the robustness of our method to poor quality data by subsampling the number of reads per gene per cell to up to 10% of the original values. We show that scVI and scANVI compares favorably to Seurat Alignment (**Appendix Figure S10**)

The strength of this paper is that they demonstrate their approach on many different datasets. Apart from the use cases provided in the manuscript by the authors, it would be useful to see if the tool can be further extended for the integration of data from different species. For example, a tool that allows for the integrative study of scRNAseq data from human and mouse. This would

be of high significance and such a tool would be a valuable resource of researchers working with model organisms.

To address this comment, we have now included an additional benchmark test for harmonizing data from humans and mice (taken from the substantia nigra; obtained from [Saunders et.al 2018] and [Welch et al., 2019]). We find consistently superior performance of scVI, using the same metrics and similar benchmark methods as in Figure 2. See new **Appendix Figure S11**.

[Saunders et.al 2018]: Saunders, Arpiar, et al. "Molecular diversity and specializations among the cells of the adult mouse brain." *Cell* 174.4 (2018): 1015-1030.

[Welch et al., 2019]: Welch, Joshua D., et al. "Single-cell multi-omic integration compares and contrasts features of brain cell identity." *Cell* 177.7 (2019): 1873-1887.

A few typos found in the document:

1. "While we demonstrate that scVI performs well in these scenarios, we also demonstrate that the latent space learned by scANVI provides a proper harmonized representation of the input datasets -" Do they mean "learned"
2. In Figure 2, the word "trajectory" is misspelled in 2 places.

We have fixed those typos

Thank you for sending us your revised manuscript. We have now heard back from the reviewer who agreed to evaluate your manuscript. You will see from the comments below that Reviewer #2 thinks that while the majority of the concerns raised by both reviewers have been addressed, the concern raised during the first round of review regarding the application of the proposed methods to analyze tumor data still remains unaddressed. In light of the reviewer comment, we would encourage you to address this remaining concern and we think that it will indeed strengthen the manuscript. However, this is not mandatory for acceptance.

On a more editorial level, please do the following.

REFEREE REPORTS

Reviewer #2:

The authors have done an admirable job in addressing our concerns, as well as those of Reviewer 1. In response to our third comment, the authors highlight their previous analyses and include a new simulation; however, they did not implement our suggestion to study tumor data. We leave it to the editor to decide how important that is for inclusion in the final version. Tumor data is substantially more difficult to analyze since cancer cells must be first delineated and their transcriptomic changes assayed according to gradation rather than easily demarcated separations. A demonstration of harmonization and annotation on single-cell tumor data would have a high impact for the field.

Reviewer #2:

The authors have done an admirable job in addressing our concerns, as well as those of Reviewer 1. In response to our third comment, the authors highlight their previous analyses and include a new simulation; however, they did not implement our suggestion to study tumor data. We leave it to the editor to decide how important that is for inclusion in the final version. Tumor data is substantially more difficult to analyze since cancer cells must be first delineated and their transcriptomic changes assayed according to gradation rather than easily demarcated separations. A demonstration of harmonization and annotation on single-cell tumor data would have a high impact for the field.

Reviewer #2 had a remaining concern regarding their suggestion to study tumor data, as an example of challenging harmonization, with the algorithm presented in the manuscript. For several reasons, we have decided to not add this additional experiment. First, we did include examples of challenging harmonization in the manuscript, including the dataset of T cells, as well as a simulated dataset with Symsim. Both are examples of transcriptomics measurement with transcriptional gradation rather than easily demarcated separation. To further emphasize this point, we added a sentence at the end of the “Cell Type Annotation in a single dataset based on “seed” labels” on page 10 to draw the connection between these datasets to other datasets with continuous variation. Second, we would like to note that scVI has been applied to cancer cell line scRNAseq data in other benchmarking studies such as [Abdelaal et al. 2019] and have been used to compare to new methods such as scAlign in [Johansen et al. 2019]. Although we concede that cancer cell-lines variations are discrete rather than continuous, these papers do demonstrate the willingness of the scientific community to use our methods, and suggest that our methods may perform well on cancer scRNAseq data.

Reference:

Abdelaal, T., Michielsen, L., Cats, D., Hoogduin, D., Mei, H., Reinders, M. J., & Mahfouz, A. (2019). A comparison of automatic cell identification methods for single-cell RNA sequencing data. *Genome biology*, 20(1), 194.

Johansen, N., & Quon, G. (2019). scAlign: a tool for alignment, integration, and rare cell identification from scRNA-seq data. *Genome biology*, 20(1), 1-21.

Thank you again for sending us your revised manuscript. We are now satisfied with the modifications made and I am pleased to inform you that your paper has been accepted for publication.

Corresponding Author Name: Nir Yosef

Manuscript Number: MSB-20-9620R